# PROGRESSIVE GROWING OF GANS FOR IMPROVED QUALITY, STABILITY, AND VARIATION

**Tero Karras**  **Timo Aila**  **Samuli Laine**  **Jaakko Lehtinen**
NVIDIA  NVIDIA  NVIDIA  NVIDIA and Aalto University
{tkarras,taila,slaine,jlehtinen}@nvidia.com

## ABSTRACT

We describe a new training methodology for generative adversarial networks. The key idea is to grow both the generator and discriminator progressively: starting from a low resolution, we add new layers that model increasingly fine details as training progresses. This both speeds the training up and greatly stabilizes it, allowing us to produce images of unprecedented quality, e.g., CELEBA images at $1024^2$. We also propose a simple way to increase the variation in generated images, and achieve a record inception score of $8.80$ in unsupervised CIFAR10. Additionally, we describe several implementation details that are important for discouraging unhealthy competition between the generator and discriminator. Finally, we suggest a new metric for evaluating GAN results, both in terms of image quality and variation. As an additional contribution, we construct a higher-quality version of the CELEBA dataset.

## 1 INTRODUCTION

Generative methods that produce novel samples from high-dimensional data distributions, such as images, are finding widespread use, for example in speech synthesis (van den Oord et al., 2016a), image-to-image translation (Zhu et al., 2017; Liu et al., 2017; Wang et al., 2017), and image inpainting (Iizuka et al., 2017). Currently the most prominent approaches are autoregressive models (van den Oord et al., 2016b;c), variational autoencoders (VAE) (Kingma & Welling, 2014), and generative adversarial networks (GAN) (Goodfellow et al., 2014). Currently they all have significant strengths and weaknesses. Autoregressive models – such as PixelCNN – produce sharp images but are slow to evaluate and do not have a latent representation as they directly model the conditional distribution over pixels, potentially limiting their applicability. VAEs are easy to train but tend to produce blurry results due to restrictions in the model, although recent work is improving this (Kingma et al., 2016). GANs produce sharp images, albeit only in fairly small resolutions and with somewhat limited variation, and the training continues to be unstable despite recent progress (Salimans et al., 2016; Gulrajani et al., 2017; Berthelot et al., 2017; Kodali et al., 2017). Hybrid methods combine various strengths of the three, but so far lag behind GANs in image quality (Makhzani & Frey, 2017; Ulyanov et al., 2017; Dumoulin et al., 2016).

Typically, a GAN consists of two networks: generator and discriminator (aka critic). The generator produces a sample, e.g., an image, from a latent code, and the distribution of these images should ideally be indistinguishable from the training distribution. Since it is generally infeasible to engineer a function that tells whether that is the case, a discriminator network is trained to do the assessment, and since networks are differentiable, we also get a gradient we can use to steer both networks to the right direction. Typically, the generator is of main interest – the discriminator is an adaptive loss function that gets discarded once the generator has been trained.

There are multiple potential problems with this formulation. When we measure the distance between the training distribution and the generated distribution, the gradients can point to more or less random directions if the distributions do not have substantial overlap, i.e., are too easy to tell apart (Arjovsky & Bottou, 2017). Originally, Jensen-Shannon divergence was used as a distance metric (Goodfellow et al., 2014), and recently that formulation has been improved (Hjelm et al., 2017) and a number of more stable alternatives have been proposed, including least squares (Mao et al., 2016b), absolute deviation with margin (Zhao et al., 2017), and Wasserstein distance (Arjovsky et al., 2017; Gulrajani

et al., 2017). Our contributions are largely orthogonal to this ongoing discussion, and we primarily use the improved Wasserstein loss, but also experiment with least-squares loss.

The generation of high-resolution images is difficult because higher resolution makes it easier to tell the generated images apart from training images (Odena et al., 2017), thus drastically amplifying the gradient problem. Large resolutions also necessitate using smaller minibatches due to memory constraints, further compromising training stability. Our key insight is that we can grow both the generator and discriminator progressively, starting from easier low-resolution images, and add new layers that introduce higher-resolution details as the training progresses. This greatly speeds up training and improves stability in high resolutions, as we will discuss in Section 2.

The GAN formulation does not explicitly require the entire training data distribution to be represented by the resulting generative model. The conventional wisdom has been that there is a tradeoff between image quality and variation, but that view has been recently challenged (Odena et al., 2017). The degree of preserved variation is currently receiving attention and various methods have been suggested for measuring it, including inception score (Salimans et al., 2016), multi-scale structural similarity (MS-SSIM) (Odena et al., 2017; Wang et al., 2003), birthday paradox (Arora & Zhang, 2017), and explicit tests for the number of discrete modes discovered (Metz et al., 2016). We will describe our method for encouraging variation in Section 3, and propose a new metric for evaluating the quality and variation in Section 5.

Section 4.1 discusses a subtle modification to the initialization of networks, leading to a more balanced learning speed for different layers. Furthermore, we observe that mode collapses traditionally plaguing GANs tend to happen very quickly, over the course of a dozen minibatches. Commonly they start when the discriminator overshoots, leading to exaggerated gradients, and an unhealthy competition follows where the signal magnitudes escalate in both networks. We propose a mechanism to stop the generator from participating in such escalation, overcoming the issue (Section 4.2).

We evaluate our contributions using the CELEBA, LSUN, CIFAR10 datasets. We improve the best published inception score for CIFAR10. Since the datasets commonly used in benchmarking generative methods are limited to a fairly low resolution, we have also created a higher quality version of the CELEBA dataset that allows experimentation with output resolutions up to $1024 \times 1024$ pixels. This dataset and our full implementation are available at `https://github.com/tkarras/progressive_growing_of_gans`, trained networks can be found at `https://drive.google.com/open?id=0B4qLcYyJmiz0NHFULTdYc05lX0U` along with result images, and a supplementary video illustrating the datasets, additional results, and latent space interpolations is at `https://youtu.be/G06dEcZ-QTg`.

## 2 PROGRESSIVE GROWING OF GANS

Our primary contribution is a training methodology for GANs where we start with low-resolution images, and then progressively increase the resolution by adding layers to the networks as visualized in Figure 1. This incremental nature allows the training to first discover large-scale structure of the image distribution and then shift attention to increasingly finer scale detail, instead of having to learn all scales simultaneously.

We use generator and discriminator networks that are mirror images of each other and always grow in synchrony. All existing layers in both networks remain trainable throughout the training process. When new layers are added to the networks, we fade them in smoothly, as illustrated in Figure 2. This avoids sudden shocks to the already well-trained, smaller-resolution layers. Appendix A describes structure of the generator and discriminator in detail, along with other training parameters.

We observe that the progressive training has several benefits. Early on, the generation of smaller images is substantially more stable because there is less class information and fewer modes (Odena et al., 2017). By increasing the resolution little by little we are continuously asking a much simpler question compared to the end goal of discovering a mapping from latent vectors to e.g. $1024^2$ images. This approach has conceptual similarity to recent work by Chen & Koltun (2017). In practice it stabilizes the training sufficiently for us to reliably synthesize megapixel-scale images using WGAN-GP loss (Gulrajani et al., 2017) and even LSGAN loss (Mao et al., 2016b).

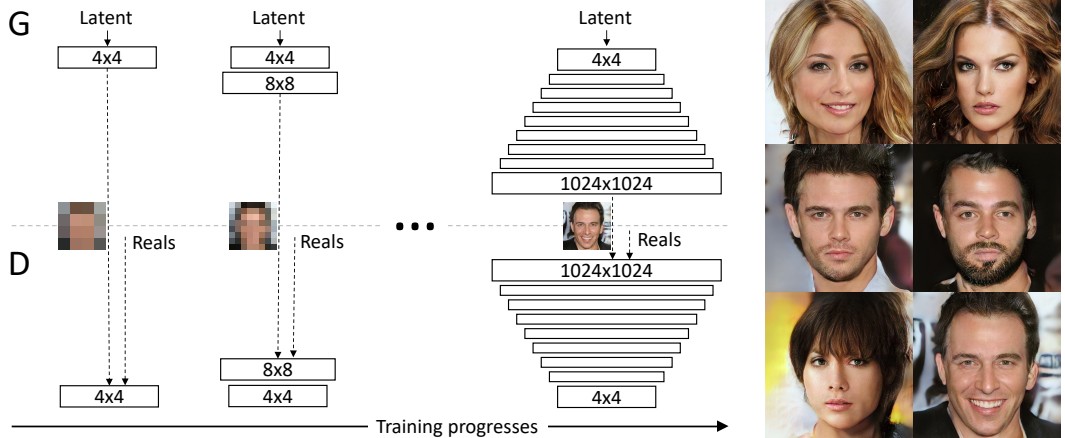

Figure 1: Our training starts with both the generator (G) and discriminator (D) having a low spatial resolution of 4×4 pixels. As the training advances, we incrementally add layers to G and D, thus increasing the spatial resolution of the generated images. All existing layers remain trainable throughout the process. Here $N \times N$ refers to convolutional layers operating on $N \times N$ spatial resolution. This allows stable synthesis in high resolutions and also speeds up training considerably. One the right we show six example images generated using progressive growing at $1024 \times 1024$.

Another benefit is the reduced training time. With progressively growing GANs most of the iterations are done at lower resolutions, and comparable result quality is often obtained up to 2–6 times faster, depending on the final output resolution.

The idea of growing GANs progressively is related to the work of Wang et al. (2017), who use multiple discriminators that operate on different spatial resolutions. That work in turn is motivated by Durugkar et al. (2016) who use one generator and multiple discriminators concurrently, and Ghosh et al. (2017) who do the opposite with multiple generators and one discriminator. Hierarchical GANs (Denton et al., 2015; Huang et al., 2016; Zhang et al., 2017) define a generator and discriminator for each level of an image pyramid. These methods build on the same observation as our work – that the complex mapping from latents to high-resolution images is easier to learn in steps – but the crucial difference is that we have only a single GAN instead of a hierarchy of them. In contrast to early work on adaptively growing networks, e.g., growing neural gas (Fritzke, 1995) and neuro evolution of augmenting topologies (Stanley & Miikkulainen, 2002) that grow networks greedily, we simply defer the introduction of pre-configured layers. In that sense our approach resembles layer-wise training of autoencoders (Bengio et al., 2007).

## 3 INCREASING VARIATION USING MINIBATCH STANDARD DEVIATION

GANs have a tendency to capture only a subset of the variation found in training data, and Salimans et al. (2016) suggest "minibatch discrimination" as a solution. They compute feature statistics not only from individual images but also across the minibatch, thus encouraging the minibatches of generated and training images to show similar statistics. This is implemented by adding a minibatch layer towards the end of the discriminator, where the layer learns a large tensor that projects the input activation to an array of statistics. A separate set of statistics is produced for each example in a minibatch and it is concatenated to the layer's output, so that the discriminator can use the statistics internally. We simplify this approach drastically while also improving the variation.

Our simplified solution has neither learnable parameters nor new hyperparameters. We first compute the standard deviation for each feature in each spatial location over the minibatch. We then average these estimates over all features and spatial locations to arrive at a single value. We replicate the value and concatenate it to all spatial locations and over the minibatch, yielding one additional (constant) feature map. This layer could be inserted anywhere in the discriminator, but we have found it best to insert it towards the end (see Appendix A.1 for details). We experimented with a richer set of statistics, but were not able to improve the variation further. In parallel work, Lin et al. (2017) provide theoretical insights about the benefits of showing multiple images to the discriminator.

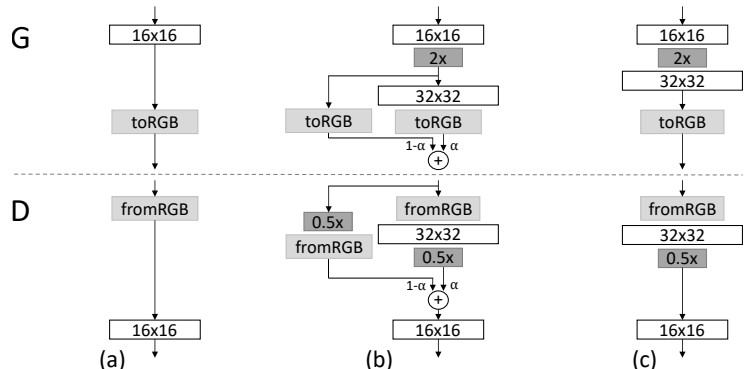

Figure 2: When doubling the resolution of the generator (G) and discriminator (D) we fade in the new layers smoothly. This example illustrates the transition from $16 \times 16$ images (a) to $32 \times 32$ images (c). During the transition (b) we treat the layers that operate on the higher resolution like a residual block, whose weight $\alpha$ increases linearly from $0$ to $1$. Here $\boxed{2\times}$ and $\boxed{0.5\times}$ refer to doubling and halving the image resolution using nearest neighbor filtering and average pooling, respectively. The $\boxed{\text{toRGB}}$ represents a layer that projects feature vectors to RGB colors and $\boxed{\text{fromRGB}}$ does the reverse; both use $1 \times 1$ convolutions. When training the discriminator, we feed in real images that are downscaled to match the current resolution of the network. During a resolution transition, we interpolate between two resolutions of the real images, similarly to how the generator output combines two resolutions.

Alternative solutions to the variation problem include unrolling the discriminator (Metz et al., 2016) to regularize its updates, and a "repelling regularizer" (Zhao et al., 2017) that adds a new loss term to the generator, trying to encourage it to orthogonalize the feature vectors in a minibatch. The multiple generators of Ghosh et al. (2017) also serve a similar goal. We acknowledge that these solutions may increase the variation even more than our solution – or possibly be orthogonal to it – but leave a detailed comparison to a later time.

## 4 NORMALIZATION IN GENERATOR AND DISCRIMINATOR

GANs are prone to the escalation of signal magnitudes as a result of unhealthy competition between the two networks. Most if not all earlier solutions discourage this by using a variant of batch normalization (Ioffe & Szegedy, 2015; Salimans & Kingma, 2016; Ba et al., 2016) in the generator, and often also in the discriminator. These normalization methods were originally introduced to eliminate covariate shift. However, we have not observed that to be an issue in GANs, and thus believe that the actual need in GANs is constraining signal magnitudes and competition. We use a different approach that consists of two ingredients, neither of which include learnable parameters.

### 4.1 EQUALIZED LEARNING RATE

We deviate from the current trend of careful weight initialization, and instead use a trivial $\mathcal{N}(0,1)$ initialization and then explicitly scale the weights at runtime. To be precise, we set $\hat{w}_i = w_i/c$, where $w_i$ are the weights and $c$ is the per-layer normalization constant from He's initializer (He et al., 2015). The benefit of doing this dynamically instead of during initialization is somewhat subtle, and relates to the scale-invariance in commonly used adaptive stochastic gradient descent methods such as RMSProp (Tieleman & Hinton, 2012) and Adam (Kingma & Ba, 2015). These methods normalize a gradient update by its estimated standard deviation, thus making the update independent of the scale of the parameter. As a result, if some parameters have a larger dynamic range than others, they will take longer to adjust. This is a scenario modern initializers cause, and thus it is possible that a learning rate is both too large and too small at the same time. Our approach ensures that the dynamic range, and thus the learning speed, is the same for all weights. A similar reasoning was independently used by van Laarhoven (2017).

## 4.2 PIXELWISE FEATURE VECTOR NORMALIZATION IN GENERATOR

To disallow the scenario where the magnitudes in the generator and discriminator spiral out of control as a result of competition, we normalize the feature vector in each pixel to unit length in the generator after each convolutional layer. We do this using a variant of "local response normalization" (Krizhevsky et al., 2012), configured as $b_{x,y} = a_{x,y}/\sqrt{\frac{1}{N}\sum_{j=0}^{N-1}(a_{x,y}^j)^2 + \epsilon}$, where $\epsilon = 10^{-8}$, $N$ is the number of feature maps, and $a_{x,y}$ and $b_{x,y}$ are the original and normalized feature vector in pixel $(x, y)$, respectively. We find it surprising that this heavy-handed constraint does not seem to harm the generator in any way, and indeed with most datasets it does not change the results much, but it prevents the escalation of signal magnitudes very effectively when needed.

## 5 MULTI-SCALE STATISTICAL SIMILARITY FOR ASSESSING GAN RESULTS

In order to compare the results of one GAN to another, one needs to investigate a large number of images, which can be tedious, difficult, and subjective. Thus it is desirable to rely on automated methods that compute some indicative metric from large image collections. We noticed that existing methods such as MS-SSIM (Odena et al., 2017) find large-scale mode collapses reliably but fail to react to smaller effects such as loss of variation in colors or textures, and they also do not directly assess image quality in terms of similarity to the training set.

We build on the intuition that a successful generator will produce samples whose local image structure is similar to the training set over all scales. We propose to study this by considering the multiscale statistical similarity between distributions of local image patches drawn from Laplacian pyramid (Burt & Adelson, 1987) representations of generated and target images, starting at a low-pass resolution of $16 \times 16$ pixels. As per standard practice, the pyramid progressively doubles until the full resolution is reached, each successive level encoding the difference to an up-sampled version of the previous level.

A single Laplacian pyramid level corresponds to a specific spatial frequency band. We randomly sample 16384 images and extract 128 descriptors from each level in the Laplacian pyramid, giving us $2^{21}$ (2.1M) descriptors per level. Each descriptor is a $7 \times 7$ pixel neighborhood with 3 color channels, denoted by $\boldsymbol{x} \in \mathbb{R}^{7 \times 7 \times 3} = \mathbb{R}^{147}$. We denote the patches from level $l$ of the training set and generated set as $\{\boldsymbol{x}_i^l\}_{i=1}^{2^{21}}$ and $\{\boldsymbol{y}_i^l\}_{i=1}^{2^{21}}$, respectively. We first normalize $\{\boldsymbol{x}_i^l\}$ and $\{\boldsymbol{y}_i^l\}$ w.r.t. the mean and standard deviation of each color channel, and then estimate the statistical similarity by computing their sliced Wasserstein distance $\mathrm{SWD}(\{\boldsymbol{x}_i^l\}, \{\boldsymbol{y}_i^l\})$, an efficiently computable randomized approximation to earthmovers distance, using 512 projections (Rabin et al., 2011).

Intuitively a small Wasserstein distance indicates that the distribution of the patches is similar, meaning that the training images and generator samples appear similar in both appearance and variation at this spatial resolution. In particular, the distance between the patch sets extracted from the lowest-resolution $16 \times 16$ images indicate similarity in large-scale image structures, while the finest-level patches encode information about pixel-level attributes such as sharpness of edges and noise.

## 6 EXPERIMENTS

In this section we discuss a set of experiments that we conducted to evaluate the quality of our results. Please refer to Appendix A for detailed description of our network structures and training configurations. We also invite the reader to consult the accompanying video (`https://youtu.be/G06dEcZ-QTg`) for additional result images and latent space interpolations. In this section we will distinguish between the *network structure* (e.g., convolutional layers, resizing), *training configuration* (various normalization layers, minibatch-related operations), and *training loss* (WGAN-GP, LSGAN).

### 6.1 IMPORTANCE OF INDIVIDUAL CONTRIBUTIONS IN TERMS OF STATISTICAL SIMILARITY

We will first use the sliced Wasserstein distance (SWD) and multi-scale structural similarity (MS-SSIM) (Odena et al., 2017) to evaluate the importance our individual contributions, and also perceptually validate the metrics themselves. We will do this by building on top of a previous state-of-the-art loss function (WGAN-GP) and training configuration (Gulrajani et al., 2017) in an unsupervised setting using CELEBA (Liu et al., 2015) and LSUN BEDROOM (Yu et al., 2015) datasets in $128^2$

| Training configuration | CELEBA | | | | | | LSUN BEDROOM | | | | | |
|---|---|---|---|---|---|---|---|---|---|---|---|---|
| | Sliced Wasserstein distance $\times 10^3$ | | | | | MS-SSIM | Sliced Wasserstein distance $\times 10^3$ | | | | | MS-SSIM |
| | 128 | 64 | 32 | 16 | Avg | | 128 | 64 | 32 | 16 | Avg | |
| (a)  Gulrajani et al. (2017) | 12.99 | 7.79 | 7.62 | 8.73 | 9.28 | 0.2854 | 11.97 | 10.51 | 8.03 | 14.48 | 11.25 | **0.0587** |
| (b)  + Progressive growing | 4.62 | **2.64** | 3.78 | 6.06 | 4.28 | **0.2838** | 7.09 | 6.27 | 7.40 | 9.64 | 7.60 | 0.0615 |
| (c)  + Small minibatch | 75.42 | 41.33 | 41.62 | 26.57 | 46.23 | 0.4065 | 72.73 | 40.16 | 42.75 | 42.46 | 49.52 | 0.1061 |
| (d)  + Revised training parameters | 9.20 | 6.53 | 4.71 | 11.84 | 8.07 | 0.3027 | 7.39 | 5.51 | 3.65 | 9.63 | 6.54 | 0.0662 |
| (e*) + Minibatch discrimination | 10.76 | 6.28 | 6.04 | 16.29 | 9.84 | 0.3057 | 10.29 | 6.22 | 5.32 | 11.88 | 8.43 | 0.0648 |
| (e)     Minibatch stddev | 13.94 | 5.67 | 2.82 | 5.71 | 7.04 | 0.2950 | 7.77 | 5.23 | 3.27 | 9.64 | 6.48 | 0.0671 |
| (f)  + Equalized learning rate | 4.42 | 3.28 | 2.32 | 7.52 | 4.39 | 0.2902 | **3.61** | 3.32 | **2.71** | 6.44 | 4.02 | 0.0668 |
| (g)  + Pixelwise normalization | **4.06** | 3.04 | **2.02** | **5.13** | **3.56** | 0.2845 | 3.89 | **3.05** | 3.24 | **5.87** | **4.01** | 0.0640 |
| (h)  Converged | 2.42 | 2.17 | 2.24 | 4.99 | 2.96 | 0.2828 | 3.47 | 2.60 | 2.30 | 4.87 | 3.31 | 0.0636 |

Table 1: Sliced Wasserstein distance (SWD) between the generated and training images (Section 5) and multi-scale structural similarity (MS-SSIM) among the generated images for several training setups at $128 \times 128$. For SWD, each column represents one level of the Laplacian pyramid, and the last one gives an average of the four distances.

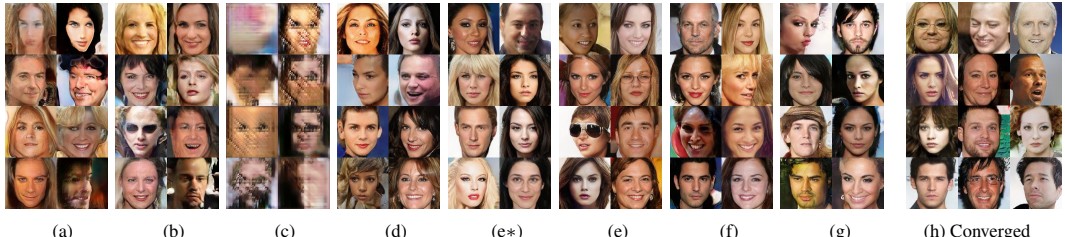

| (a) | (b) | (c) | (d) | (e∗) | (e) | (f) | (g) | (h) Converged |

Figure 3: (a) – (g) CELEBA examples corresponding to rows in Table 1. These are intentionally non-converged. (h) Our converged result. Notice that some images show aliasing and some are not sharp – this is a flaw of the dataset, which the model learns to replicate faithfully.

resolution. CELEBA is particularly well suited for such comparison because the training images contain noticeable artifacts (aliasing, compression, blur) that are difficult for the generator to reproduce faithfully. In this test we amplify the differences between training configurations by choosing a relatively low-capacity network structure (Appendix A.2) and terminating the training once the discriminator has been shown a total of 10M real images. As such the results are not fully converged.

Table 1 lists the numerical values for SWD and MS-SSIM in several training configurations, where our individual contributions are cumulatively enabled one by one on top of the baseline (Gulrajani et al., 2017). The MS-SSIM numbers were averaged from 10000 pairs of generated images, and SWD was calculated as described in Section 5. Generated CELEBA images from these configurations are shown in Figure 3. Due to space constraints, the figure shows only a small number of examples for each row of the table, but a significantly broader set is available in Appendix H. Intuitively, a good evaluation metric should reward plausible images that exhibit plenty of variation in colors, textures, and viewpoints. However, this is not captured by MS-SSIM: we can immediately see that configuration (h) generates significantly better images than configuration (a), but MS-SSIM remains approximately unchanged because it measures only the variation between outputs, not similarity to the training set. SWD, on the other hand, does indicate a clear improvement.

The first training configuration (a) corresponds to Gulrajani et al. (2017), featuring batch normalization in the generator, layer normalization in the discriminator, and minibatch size of 64. (b) enables progressive growing of the networks, which results in sharper and more believable output images. SWD correctly finds the distribution of generated images to be more similar to the training set.

Our primary goal is to enable high output resolutions, and this requires reducing the size of minibatches in order to stay within the available memory budget. We illustrate the ensuing challenges in (c) where we decrease the minibatch size from 64 to 16. The generated images are unnatural, which is clearly visible in both metrics. In (d), we stabilize the training process by adjusting the hyperparameters as well as by removing batch normalization and layer normalization (Appendix A.2). As an intermediate test (e∗), we enable minibatch discrimination (Salimans et al., 2016), which somewhat surprisingly fails to improve any of the metrics, including MS-SSIM that measures output variation. In contrast, our minibatch standard deviation (e) improves the average SWD scores and images. We then enable our remaining contributions in (f) and (g), leading to an overall improvement in SWD

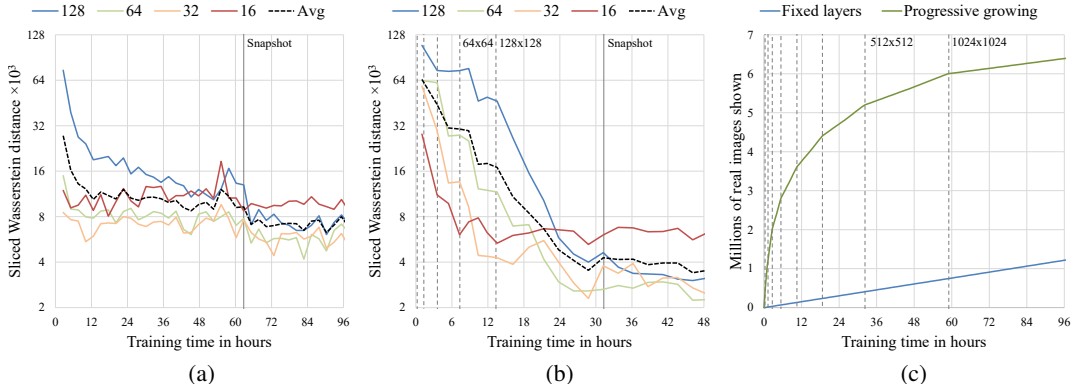

Figure 4: Effect of progressive growing on training speed and convergence. The timings were measured on a single-GPU setup using NVIDIA Tesla P100. (a) Statistical similarity with respect to wall clock time for Gulrajani et al. (2017) using CELEBA at $128 \times 128$ resolution. Each graph represents sliced Wasserstein distance on one level of the Laplacian pyramid, and the vertical line indicates the point where we stop the training in Table 1. (b) Same graph with progressive growing enabled. The dashed vertical lines indicate points where we double the resolution of G and D. (c) Effect of progressive growing on the raw training speed in $1024 \times 1024$ resolution.

and subjective visual quality. Finally, in (h) we use a non-crippled network and longer training – we feel the quality of the generated images is at least comparable to the best published results so far.

## 6.2    CONVERGENCE AND TRAINING SPEED

Figure 4 illustrates the effect of progressive growing in terms of the SWD metric and raw image throughput. The first two plots correspond to the training configuration of Gulrajani et al. (2017) without and with progressive growing. We observe that the progressive variant offers two main benefits: it converges to a considerably better optimum and also reduces the total training time by about a factor of two. The improved convergence is explained by an implicit form of curriculum learning that is imposed by the gradually increasing network capacity. Without progressive growing, all layers of the generator and discriminator are tasked with simultaneously finding succinct intermediate representations for both the large-scale variation and the small-scale detail. With progressive growing, however, the existing low-resolution layers are likely to have already converged early on, so the networks are only tasked with refining the representations by increasingly smaller-scale effects as new layers are introduced. Indeed, we see in Figure 4(b) that the largest-scale statistical similarity curve (16) reaches its optimal value very quickly and remains consistent throughout the rest of the training. The smaller-scale curves (32, 64, 128) level off one by one as the resolution is increased, but the convergence of each curve is equally consistent. With non-progressive training in Figure 4(a), each scale of the SWD metric converges roughly in unison, as could be expected.

The speedup from progressive growing increases as the output resolution grows. Figure 4(c) shows training progress, measured in number of real images shown to the discriminator, as a function of training time when the training progresses all the way to $1024^2$ resolution. We see that progressive growing gains a significant head start because the networks are shallow and quick to evaluate at the beginning. Once the full resolution is reached, the image throughput is equal between the two methods. The plot shows that the progressive variant reaches approximately 6.4 million images in 96 hours, whereas it can be extrapolated that the non-progressive variant would take about 520 hours to reach the same point. In this case, the progressive growing offers roughly a $5.4\times$ speedup.

## 6.3    HIGH-RESOLUTION IMAGE GENERATION USING CELEBA-HQ DATASET

To meaningfully demonstrate our results at high output resolutions, we need a sufficiently varied high-quality dataset. However, virtually all publicly available datasets previously used in GAN literature are limited to relatively low resolutions ranging from $32^2$ to $480^2$. To this end, we created a high-quality version of the CELEBA dataset consisting of 30000 of the images at $1024 \times 1024$ resolution. We refer to Appendix C for further details about the generation of this dataset.

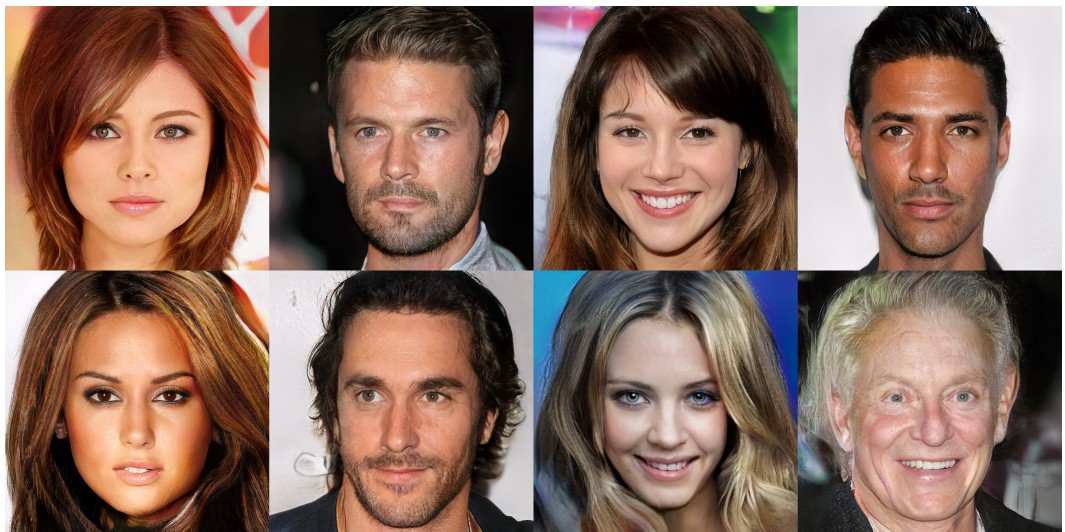

Figure 5: $1024 \times 1024$ images generated using the CELEBA-HQ dataset. See Appendix F for a larger set of results, and the accompanying video for latent space interpolations.

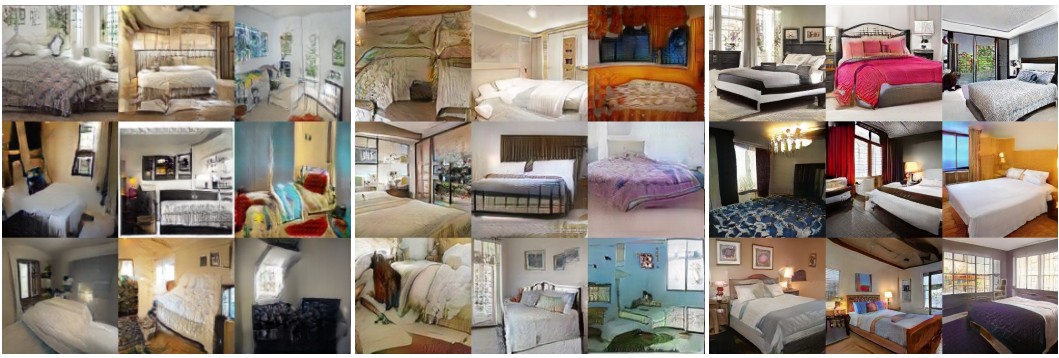

Mao et al. (2016b) ($128 \times 128$)    Gulrajani et al. (2017) ($128 \times 128$)    Our ($256 \times 256$)

Figure 6: Visual quality comparison in LSUN BEDROOM; pictures copied from the cited articles.

Our contributions allow us to deal with high output resolutions in a robust and efficient fashion. Figure 5 shows selected $1024 \times 1024$ images produced by our network. While megapixel GAN results have been shown before in another dataset (Marchesi, 2017), our results are vastly more varied and of higher perceptual quality. Please refer to Appendix F for a larger set of result images as well as the nearest neighbors found from the training data. The accompanying video shows latent space interpolations and visualizes the progressive training. The interpolation works so that we first randomize a latent code for each frame (512 components sampled individually from $\mathcal{N}(0, 1)$), then blur the latents across time with a Gaussian ($\sigma = 45$ frames @ 60Hz), and finally normalize each vector to lie on a hypersphere.

We trained the network on 8 Tesla V100 GPUs for 4 days, after which we no longer observed qualitative differences between the results of consecutive training iterations. Our implementation used an adaptive minibatch size depending on the current output resolution so that the available memory budget was optimally utilized.

In order to demonstrate that our contributions are largely orthogonal to the choice of a loss function, we also trained the same network using LSGAN loss instead of WGAN-GP loss. Figure 1 shows six examples of $1024^2$ images produced using our method using LSGAN. Further details of this setup are given in Appendix B.

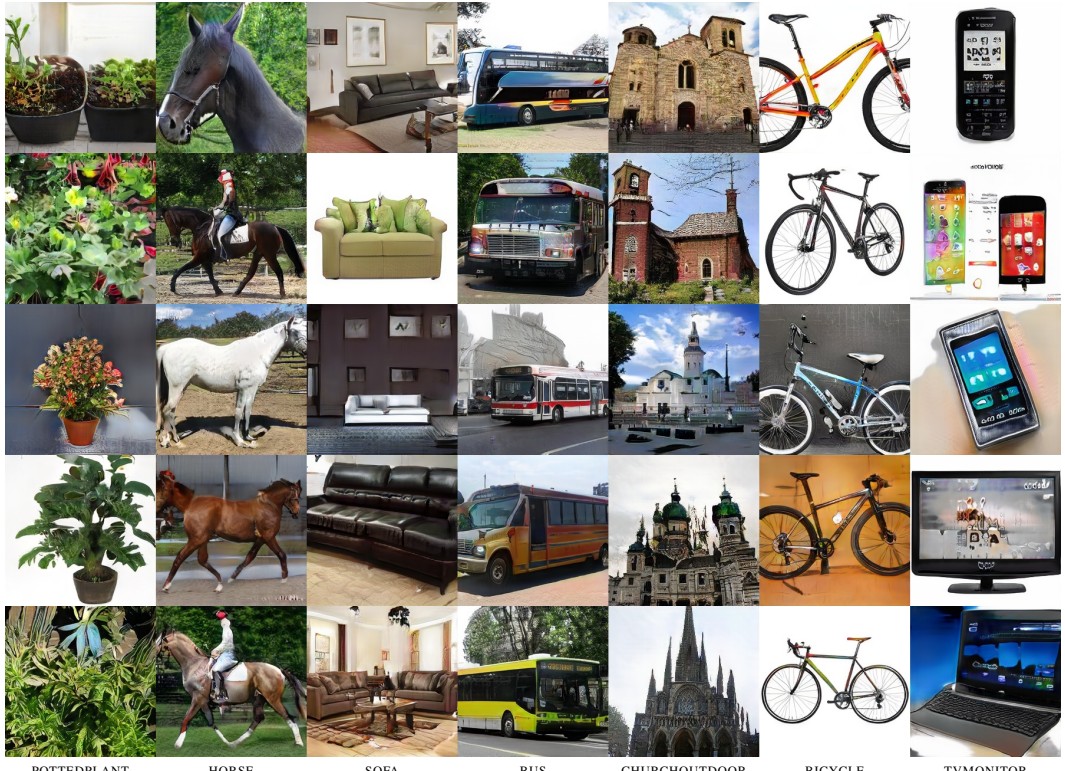

| POTTEDPLANT | HORSE | SOFA | BUS | CHURCHOUTDOOR | BICYCLE | TVMONITOR |

Figure 7: Selection of $256 \times 256$ images generated from different LSUN categories.

## 6.4 LSUN RESULTS

Figure 6 shows a purely visual comparison between our solution and earlier results in LSUN BED-ROOM. Figure 7 gives selected examples from seven very different LSUN categories at $256^2$. A larger, non-curated set of results from all 30 LSUN categories is available in Appendix G, and the video demonstrates interpolations. We are not aware of earlier results in most of these categories, and while some categories work better than others, we feel that the overall quality is high.

## 6.5 CIFAR10 INCEPTION SCORES

The best inception scores for CIFAR10 (10 categories of $32 \times 32$ RGB images) we are aware of are 7.90 for unsupervised and 8.87 for label conditioned setups (Grinblat et al., 2017). The large difference between the two numbers is primarily caused by "ghosts" that necessarily appear between classes in the unsupervised setting, while label conditioning can remove many such transitions.

When all of our contributions are enabled, we get $8.80$ in the *unsupervised* setting. Appendix D shows a representative set of generated images along with a more comprehensive list of results from earlier methods. The network and training setup were the same as for CELEBA, progression limited to $32 \times 32$ of course. The only customization was to the WGAN-GP's regularization term $\mathbb{E}_{\hat{\mathbf{x}} \sim \mathbb{P}_{\hat{\mathbf{x}}}}[(||\nabla_{\hat{\mathbf{x}}} D(\hat{\mathbf{x}})||_2 - \gamma)^2/\gamma^2]$. Gulrajani et al. (2017) used $\gamma = 1.0$, which corresponds to 1-Lipschitz, but we noticed that it is in fact significantly better to prefer fast transitions ($\gamma = 750$) to minimize the ghosts. We have not tried this trick with other datasets.

## 7 DISCUSSION

While the quality of our results is generally high compared to earlier work on GANs, and the training is stable in large resolutions, there is a long way to true photorealism. Semantic sensibility and understanding dataset-dependent constraints, such as certain objects being straight rather than curved, leaves a lot to be desired. There is also room for improvement in the micro-structure of the images. That said, we feel that convincing realism may now be within reach, especially in CELEBA-HQ.

## 8 ACKNOWLEDGEMENTS

We would like to thank Mikael Honkavaara, Tero Kuosmanen, and Timi Hietanen for the compute infrastructure. Dmitry Korobchenko and Richard Calderwood for efforts related to the CELEBA-HQ dataset. Oskar Elek, Jacob Munkberg, and Jon Hasselgren for useful comments.

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

| Generator | Act. | Output shape | Params |
|---|---|---|---|
| Latent vector | – | 512 × 1 × 1 | – |
| Conv 4 × 4 | LReLU | 512 × 4 × 4 | 4.2M |
| Conv 3 × 3 | LReLU | 512 × 4 × 4 | 2.4M |
| Upsample | – | 512 × 8 × 8 | – |
| Conv 3 × 3 | LReLU | 512 × 8 × 8 | 2.4M |
| Conv 3 × 3 | LReLU | 512 × 8 × 8 | 2.4M |
| Upsample | – | 512 × 16 × 16 | – |
| Conv 3 × 3 | LReLU | 512 × 16 × 16 | 2.4M |
| Conv 3 × 3 | LReLU | 512 × 16 × 16 | 2.4M |
| Upsample | – | 512 × 32 × 32 | – |
| Conv 3 × 3 | LReLU | 512 × 32 × 32 | 2.4M |
| Conv 3 × 3 | LReLU | 512 × 32 × 32 | 2.4M |
| Upsample | – | 512 × 64 × 64 | – |
| Conv 3 × 3 | LReLU | 256 × 64 × 64 | 1.2M |
| Conv 3 × 3 | LReLU | 256 × 64 × 64 | 590k |
| Upsample | – | 256 × 128 × 128 | – |
| Conv 3 × 3 | LReLU | 128 × 128 × 128 | 295k |
| Conv 3 × 3 | LReLU | 128 × 128 × 128 | 148k |
| Upsample | – | 128 × 256 × 256 | – |
| Conv 3 × 3 | LReLU | 64 × 256 × 256 | 74k |
| Conv 3 × 3 | LReLU | 64 × 256 × 256 | 37k |
| Upsample | – | 64 × 512 × 512 | – |
| Conv 3 × 3 | LReLU | 32 × 512 × 512 | 18k |
| Conv 3 × 3 | LReLU | 32 × 512 × 512 | 9.2k |
| Upsample | – | 32 × 1024 × 1024 | – |
| Conv 3 × 3 | LReLU | 16 × 1024 × 1024 | 4.6k |
| Conv 3 × 3 | LReLU | 16 × 1024 × 1024 | 2.3k |
| Conv 1 × 1 | linear | 3 × 1024 × 1024 | 51 |
| Total trainable parameters | | | **23.1M** |

| Discriminator | Act. | Output shape | Params |
|---|---|---|---|
| Input image | – | 3 × 1024 × 1024 | – |
| Conv 1 × 1 | LReLU | 16 × 1024 × 1024 | 64 |
| Conv 3 × 3 | LReLU | 16 × 1024 × 1024 | 2.3k |
| Conv 3 × 3 | LReLU | 32 × 1024 × 1024 | 4.6k |
| Downsample | – | 32 × 512 × 512 | – |
| Conv 3 × 3 | LReLU | 32 × 512 × 512 | 9.2k |
| Conv 3 × 3 | LReLU | 64 × 512 × 512 | 18k |
| Downsample | – | 64 × 256 × 256 | – |
| Conv 3 × 3 | LReLU | 64 × 256 × 256 | 37k |
| Conv 3 × 3 | LReLU | 128 × 256 × 256 | 74k |
| Downsample | – | 128 × 128 × 128 | – |
| Conv 3 × 3 | LReLU | 128 × 128 × 128 | 148k |
| Conv 3 × 3 | LReLU | 256 × 128 × 128 | 295k |
| Downsample | – | 256 × 64 × 64 | – |
| Conv 3 × 3 | LReLU | 256 × 64 × 64 | 590k |
| Conv 3 × 3 | LReLU | 512 × 64 × 64 | 1.2M |
| Downsample | – | 512 × 32 × 32 | – |
| Conv 3 × 3 | LReLU | 512 × 32 × 32 | 2.4M |
| Conv 3 × 3 | LReLU | 512 × 32 × 32 | 2.4M |
| Downsample | – | 512 × 16 × 16 | – |
| Conv 3 × 3 | LReLU | 512 × 16 × 16 | 2.4M |
| Conv 3 × 3 | LReLU | 512 × 16 × 16 | 2.4M |
| Downsample | – | 512 × 8 × 8 | – |
| Conv 3 × 3 | LReLU | 512 × 8 × 8 | 2.4M |
| Conv 3 × 3 | LReLU | 512 × 8 × 8 | 2.4M |
| Downsample | – | 512 × 4 × 4 | – |
| Minibatch stddev | – | 513 × 4 × 4 | – |
| Conv 3 × 3 | LReLU | 512 × 4 × 4 | 2.4M |
| Conv 4 × 4 | LReLU | 512 × 1 × 1 | 4.2M |
| Fully-connected | linear | 1 × 1 × 1 | 513 |
| Total trainable parameters | | | **23.1M** |

Table 2: Generator and discriminator that we use with CELEBA-HQ to generate 1024×1024 images.

## A  NETWORK STRUCTURE AND TRAINING CONFIGURATION

### A.1  1024 × 1024 NETWORKS USED FOR CELEBA-HQ

Table 2 shows network architectures of the full-resolution generator and discriminator that we use with the CELEBA-HQ dataset. Both networks consist mainly of replicated 3-layer blocks that we introduce one by one during the course of the training. The last Conv 1 × 1 layer of the generator corresponds to the `toRGB` block in Figure 2, and the first Conv 1 × 1 layer of the discriminator similarly corresponds to `fromRGB`. We start with 4 × 4 resolution and train the networks until we have shown the discriminator 800k real images in total. We then alternate between two phases: fade in the first 3-layer block during the next 800k images, stabilize the networks for 800k images, fade in the next 3-layer block during 800k images, etc.

Our latent vectors correspond to random points on a 512-dimensional hypersphere, and we represent training and generated images in [-1,1]. We use leaky ReLU with leakiness 0.2 in all layers of both networks, except for the last layer that uses linear activation. We do not employ batch normalization, layer normalization, or weight normalization in either network, but we perform pixelwise normalization of the feature vectors after each Conv 3 × 3 layer in the generator as described in Section 4.2. We initialize all bias parameters to zero and all weights according to the normal distribution with unit variance. However, we scale the weights with a layer-specific constant at runtime as described in Section 4.1. We inject the across-minibatch standard deviation as an additional feature map at 4 × 4 resolution toward the end of the discriminator as described in Section 3. The upsampling and downsampling operations in Table 2 correspond to 2 × 2 element replication and average pooling, respectively.

We train the networks using Adam (Kingma & Ba, 2015) with $\alpha = 0.001$, $\beta_1 = 0$, $\beta_2 = 0.99$, and $\epsilon = 10^{-8}$. We do not use any learning rate decay or rampdown, but for visualizing generator output at any given point during the training, we use an exponential running average for the weights of the generator with decay 0.999. We use a minibatch size 16 for resolutions $4^2$–$128^2$ and then gradually decrease the size according to $256^2 \rightarrow 14$, $512^2 \rightarrow 6$, and $1024^2 \rightarrow 3$ to avoid exceeding the available memory budget. We use the WGAN-GP loss, but unlike Gulrajani et al. (2017), we alternate between optimizing the generator and discriminator on a per-minibatch basis, i.e., we set $n_{\text{critic}} = 1$. Additionally, we introduce a fourth term into the discriminator loss with an extremely

small weight to keep the discriminator output from drifting too far away from zero. To be precise, we set $L' = L + \epsilon_{\text{drift}} \mathbb{E}_{x \in \mathbb{P}_r}[D(x)^2]$, where $\epsilon_{\text{drift}} = 0.001$.

## A.2 OTHER NETWORKS

Whenever we need to operate on a spatial resolution lower than $1024 \times 1024$, we do that by leaving out an appropriate number copies of the replicated 3-layer block in both networks.

Furthermore, Section 6.1 uses a slightly lower-capacity version, where we halve the number of feature maps in Conv $3 \times 3$ layers at the $16 \times 16$ resolution, and divide by 4 in the subsequent resolutions. This leaves 32 feature maps to the last Conv $3 \times 3$ layers. In Table 1 and Figure 4 we train each resolution for a total 600k images instead of 800k, and also fade in new layers for the duration of 600k images.

For the "Gulrajani et al. (2017)" case in Table 1, we follow their training configuration as closely as possible. In particular, we set $\alpha = 0.0001$, $\beta_2 = 0.9$, $n_{\text{critic}} = 5$, $\epsilon_{\text{drift}} = 0$, and minibatch size 64. We disable progressive resolution, minibatch stddev, as well as weight scaling at runtime, and initialize all weights using He's initializer (He et al., 2015). Furthermore, we modify the generator by replacing LReLU with ReLU, linear activation with tanh in the last layer, and pixelwise normalization with batch normalization. In the discriminator, we add layer normalization to all Conv $3 \times 3$ and Conv $4 \times 4$ layers. For the latent vectors, we use 128 components sampled independently from the normal distribution.

## B LEAST-SQUARES GAN (LSGAN) AT $1024 \times 1024$

We find that LSGAN is generally a less stable loss function than WGAN-GP, and it also has a tendency to lose some of the variation towards the end of long runs. Thus we prefer WGAN-GP, but have also produced high-resolution images by building on top of LSGAN. For example, the $1024^2$ images in Figure 1 are LSGAN-based.

On top of the techniques described in Sections 2–4, we need one additional hack with LSGAN that prevents the training from spiraling out of control when the dataset is too easy for the discriminator, and the discriminator gradients are at risk of becoming meaningless as a result. We adaptively increase the magnitude of multiplicative Gaussian noise in discriminator as a function of the discriminator's output. The noise is applied to the input of each Conv $3 \times 3$ and Conv $4 \times 4$ layer. There is a long history of adding noise to the discriminator, and it is generally detrimental for the image quality (Arjovsky et al., 2017) and ideally one would never have to do that, which according to our tests is the case for WGAN-GP (Gulrajani et al., 2017). The magnitude of noise is determined as $0.2 \cdot \max(0, \hat{d}_t - 0.5)^2$, where $\hat{d}_t = 0.1d + 0.9\hat{d}_{t-1}$ is an exponential moving average of the discriminator output $d$. The motivation behind this hack is that LSGAN is seriously unstable when $d$ approaches (or exceeds) 1.0.

## C CELEBA-HQ DATASET

In this section we describe the process we used to create the high-quality version of the CELEBA dataset, consisting of 30000 images in $1024 \times 1024$ resolution. As a starting point, we took the collection of in-the-wild images included as a part of the original CELEBA dataset. These images are extremely varied in terms of resolution and visual quality, ranging all the way from $43 \times 55$ to $6732 \times 8984$. Some of them show crowds of several people whereas others focus on the face of a single person – often only a part of the face. Thus, we found it necessary to apply several image processing steps to ensure consistent quality and to center the images on the facial region.

Our processing pipeline is illustrated in Figure 8. To improve the overall image quality, we pre-process each JPEG image using two pre-trained neural networks: a convolutional autoencoder trained to remove JPEG artifacts in natural images, similar in structure to the proposed by Mao et al. (2016a), and an adversarially-trained 4x super-resolution network (Korobchenko & Foco, 2017) similar to Ledig et al. (2016). To handle cases where the facial region extends outside the image, we employ padding and filtering to extend the dimensions of the image as illustrated in Fig.8(c–d). We then select an oriented crop rectangle based on the facial landmark annotations included in the

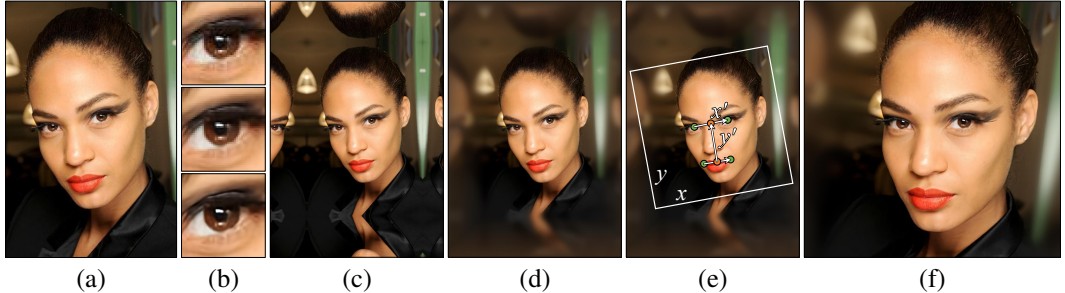

|     |     |     |     |     |     |
| --- | --- | --- | --- | --- | --- |
| (a) | (b) | (c) | (d) | (e) | (f) |

Figure 8: Creating the CELEBA-HQ dataset. We start with a JPEG image (a) from the CelebA in-the-wild dataset. We improve the visual quality (b,top) through JPEG artifact removal (b,middle) and 4x super-resolution (b,bottom). We then extend the image through mirror padding (c) and Gaussian filtering (d) to produce a visually pleasing depth-of-field effect. Finally, we use the facial landmark locations to select an appropriate crop region (e) and perform high-quality resampling to obtain the final image at $1024 \times 1024$ resolution (f).

original CELEBA dataset as follows:

$$
\begin{aligned}
x' &= e_1 - e_0 \\
y' &= \frac{1}{2}(e_0 + e_1) - \frac{1}{2}(m_0 + m_1) \\
c &= \frac{1}{2}(e_0 + e_1) - 0.1 \cdot y' \\
s &= \max\left(4.0 \cdot |x'|, 3.6 \cdot |y'|\right) \\
x &= \mathrm{Normalize}\left(x' - \mathrm{Rotate90}(y')\right) \\
y &= \mathrm{Rotate90}(x)
\end{aligned}
$$

$e_0$, $e_1$, $m_0$, and $m_1$ represent the 2D pixel locations of the two eye landmarks and two mouth landmarks, respectively, $c$ and $s$ indicate the center and size of the desired crop rectangle, and $x$ and $y$ indicate its orientation. We constructed the above formulas empirically to ensure that the crop rectangle stays consistent in cases where the face is viewed from different angles. Once we have calculated the crop rectangle, we transform the rectangle to $4096 \times 4096$ pixels using bilinear filtering, and then scale it to $1024 \times 1024$ resolution using a box filter.

We perform the above processing for all 202599 images in the dataset, analyze the resulting $1024 \times 1024$ images further to estimate the final image quality, sort the images accordingly, and discard all but the best 30000 images. We use a frequency-based quality metric that favors images whose power spectrum contains a broad range of frequencies and is approximately radially symmetric. This penalizes blurry images as well as images that have conspicuous directional features due to, e.g., visible halftoning patterns. We selected the cutoff point of 30000 images as a practical sweet spot between variation and image quality, because it appeared to yield the best results.

## D CIFAR10 RESULTS

Figure 9 shows non-curated images generated in the unsupervised setting, and Table 3 compares against prior art in terms of inception scores. We report our scores in two different ways: 1) the highest score observed during training runs (here $\pm$ refers to the standard deviation returned by the inception score calculator) and 2) the mean and standard deviation computed from the highest scores seen during training, starting from ten random initializations. Arguably the latter methodology is much more meaningful as one can be lucky with individual runs (as we were). We did not use any kind of augmentation with this dataset.

## E MNIST-1K DISCRETE MODE TEST WITH CRIPPLED DISCRIMINATOR

Metz et al. (2016) describe a setup where a generator synthesizes MNIST digits simultaneously to 3 color channels, the digits are classified using a pre-trained classifier ($0.4\%$ error rate in our case), and concatenated to form a number in $[0, 999]$. They generate a total of 25,600 images and count

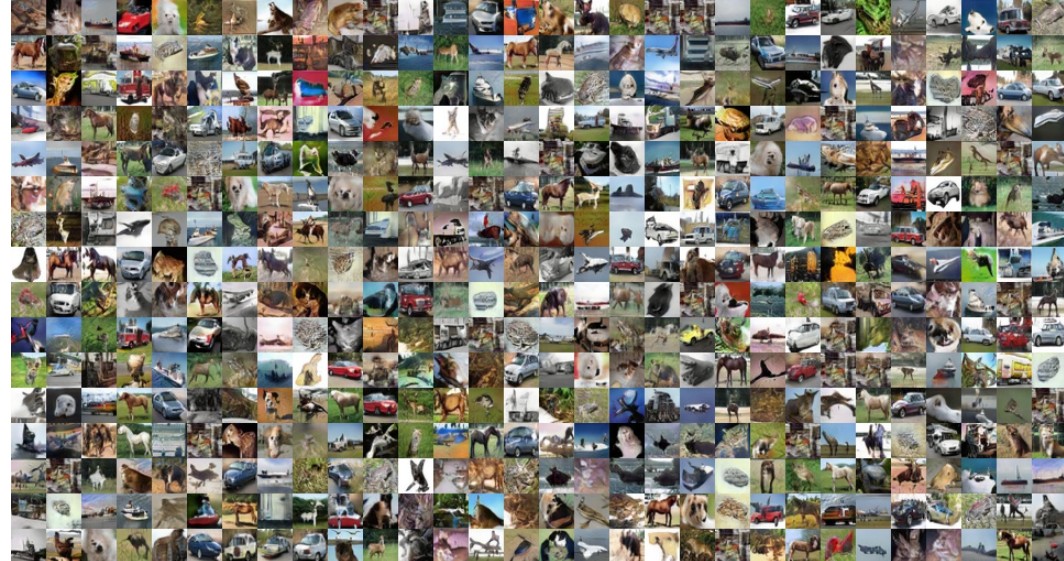

Figure 9: CIFAR10 images generated using a network that was trained unsupervised (no label conditioning), and achieves a record 8.80 inception score.

| UNSUPERVISED | | | LABEL CONDITIONED | | |
|---|---|---|---|---|---|
| Method | | Inception score | Method | | Inception score |
| ALI | (Dumoulin et al., 2016) | $5.34 \pm 0.05$ | DCGAN | (Radford et al., 2015) | 6.58 |
| GMAN | (Durugkar et al., 2016) | $6.00 \pm 0.19$ | Improved GAN | (Salimans et al., 2016) | $8.09 \pm 0.07$ |
| Improved GAN | (Salimans et al., 2016) | $6.86 \pm 0.06$ | AC-GAN | (Odena et al., 2017) | $8.25 \pm 0.07$ |
| CEGAN-Ent-VI | (Dai et al., 2017) | $7.07 \pm 0.07$ | SGAN | (Huang et al., 2016) | $8.59 \pm 0.12$ |
| LR-AGN | (Yang et al., 2017) | $7.17 \pm 0.17$ | WGAN-GP | (Gulrajani et al., 2017) | $8.67 \pm 0.14$ |
| DFM | (Warde-Farley & Bengio, 2017) | $7.72 \pm 0.13$ | Splitting GAN | (Grinblat et al., 2017) | $8.87 \pm 0.09$ |
| WGAN-GP | (Gulrajani et al., 2017) | $7.86 \pm 0.07$ | | | |
| Splitting GAN | (Grinblat et al., 2017) | $7.90 \pm 0.09$ | | | |
| Our (best run) | | $\mathbf{8.80 \pm 0.05}$ | | | |
| Our (computed from 10 runs) | | $\mathbf{8.56 \pm 0.06}$ | | | |

Table 3: CIFAR10 inception scores, higher is better.

how many of the discrete modes are covered. They also compute KL divergence as KL(histogram || uniform). Modern GAN implementations can trivially cover all modes at very low divergence (0.05 in our case), and thus Metz et al. specify a fairly low-capacity generator and two severely crippled discriminators ("K/2" has $\sim$ 2000 params and "K/4" only about $\sim$ 500) to tease out differences between training methodologies. Both of these networks use batch normalization.

As shown in Table 4, using WGAN-GP loss with the networks specified by Metz et al. covers much more modes than the original GAN loss, and even more than the unrolled original GAN with the smaller (K/4) discriminator. The KL divergence, which is arguably a more accurate metric than the raw count, acts even more favorably.

Replacing batch normalization with our normalization (equalized learning rate, pixelwise normalization) improves the result considerably, while also removing a few trainable parameters from the discriminators. The addition of a minibatch stddev layer further improves the scores, while restoring the discriminator capacity to within 0.5% of the original. Progression does not help much with these tiny images, but it does not hurt either.

| Arch | | GAN | + unrolling | WGAN-GP | + our norm | + mb stddev | + progression |
|---|---|---|---|---|---|---|---|
| K/4 | # | $30.6 \pm 20.7$ | $372.2 \pm 20.7$ | $640.1 \pm 136.3$ | $856.7 \pm 50.4$ | $\mathbf{881.3 \pm 39.2}$ | $859.5 \pm 36.2$ |
| | KL | $5.99 \pm 0.04$ | $4.66 \pm 0.46$ | $1.97 \pm 0.70$ | $1.10 \pm 0.19$ | $1.09 \pm 0.16$ | $\mathbf{1.05 \pm 0.09}$ |
| K/2 | # | $628.0 \pm 140.9$ | $817.4 \pm 39.9$ | $772.4 \pm 146.5$ | $886.6 \pm 58.5$ | $918.3 \pm 30.2$ | $\mathbf{919.8 \pm 35.1}$ |
| | KL | $2.58 \pm 0.75$ | $1.43 \pm 0.12$ | $1.35 \pm 0.55$ | $0.98 \pm 0.33$ | $0.89 \pm 0.21$ | $\mathbf{0.82 \pm 0.13}$ |

Table 4: Results for MNIST discrete mode test using two tiny discriminators (K/4, K/2) defined by Metz et al. (2016). The number of covered modes (#) and KL divergence from a uniform distribution are given as an average $\pm$ standard deviation over 8 random initializations. Higher is better for the number of modes, and lower is better for KL divergence.

## F ADDITIONAL CELEBA-HQ RESULTS

Figure 10 shows the nearest neighbors found for our generated images. Figure 11 gives additional generated examples from CELEBA-HQ. We enabled mirror augmentation for all tests using CELEBA and CELEBA-HQ. In addition to the sliced Wasserstein distance (SWD), we also quote the recently introduced Fréchet Inception Distance (FID) (Heusel et al., 2017) computed from 50K images.

## G LSUN RESULTS

Figures 12–17 show representative images generated for all 30 LSUN categories. A separate network was trained for each category using identical parameters. All categories were trained using 100k images, except for BEDROOM and DOG that used all the available data. Since 100k images is a very limited amount of training data for most categories, we enabled mirror augmentation in these tests (but not for BEDROOM or DOG).

## H ADDITIONAL IMAGES FOR TABLE 1

Figure 18 shows larger collections of images corresponding to the non-converged setups in Table 1. The training time was intentionally limited to make the differences between various methods more visible.

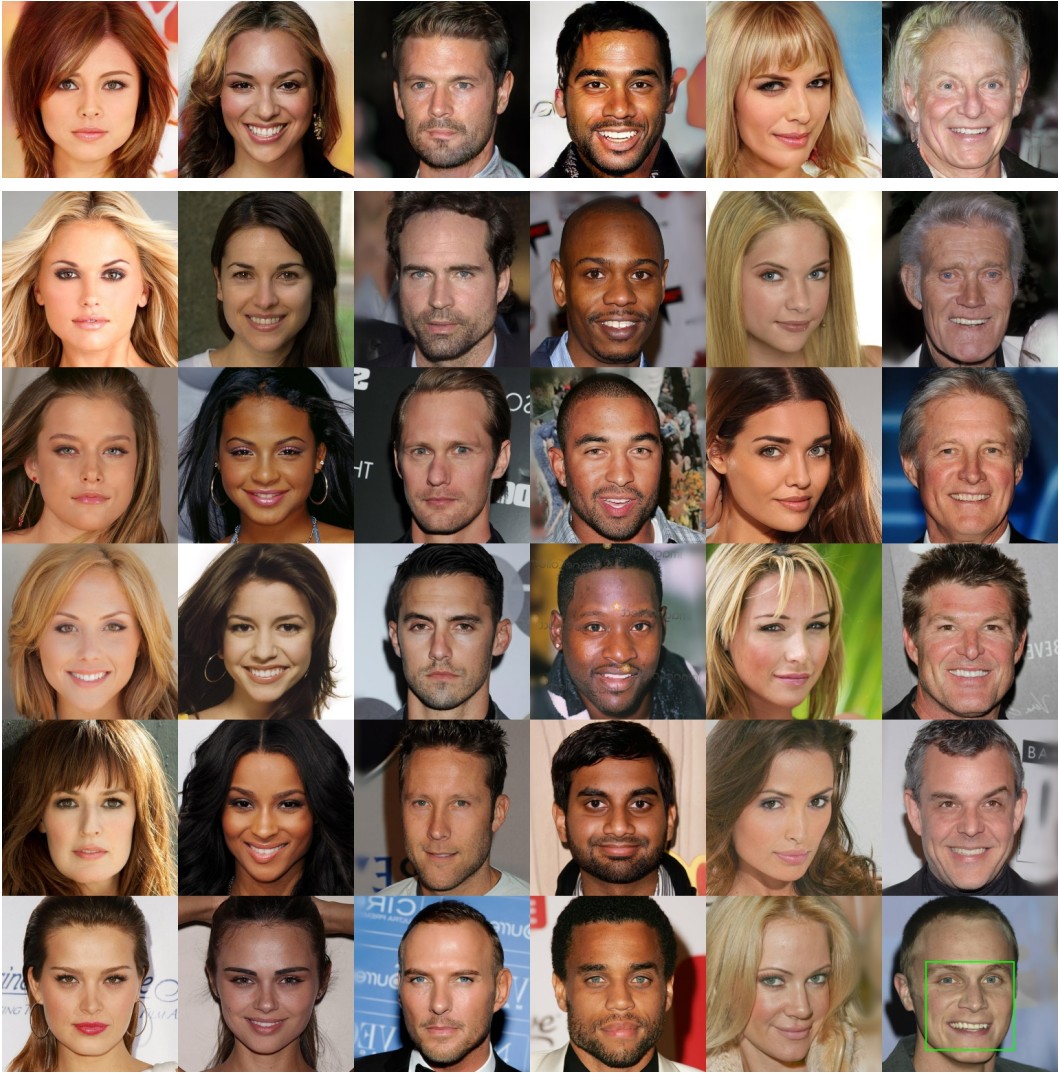

Figure 10: Top: Our CELEBA-HQ results. Next five rows: Nearest neighbors found from the training data, based on feature-space distance. We used activations from five VGG layers, as suggested by Chen & Koltun (2017). Only the crop highlighted in bottom right image was used for comparison in order to exclude image background and focus the search on matching facial features.

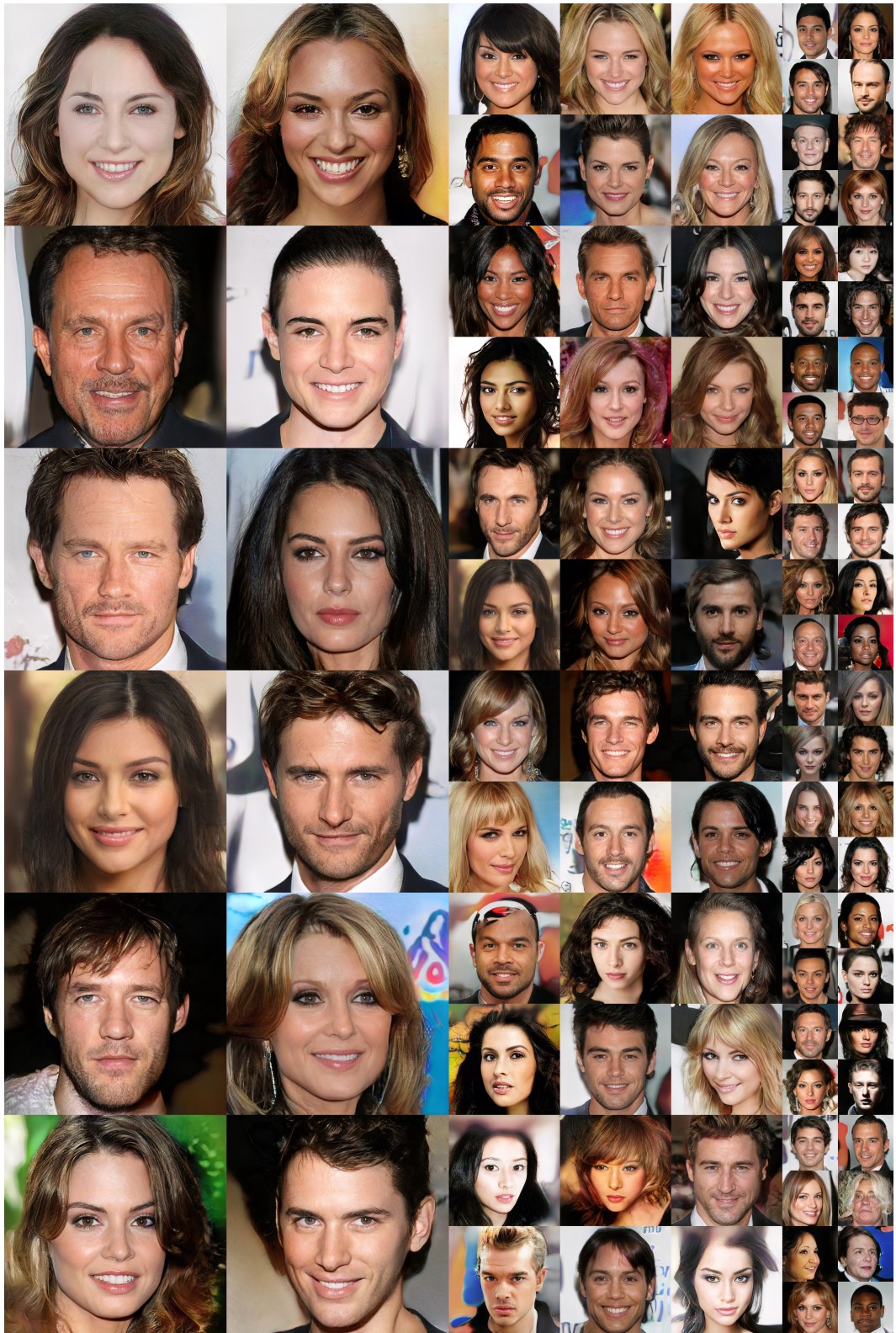

Figure 11: Additional $1024 \times 1024$ images generated using the CELEBA-HQ dataset. Sliced Wasserstein Distance (SWD) $\times 10^3$ for levels 1024, . . . , 16: 7.48, 7.24, 6.08, 3.51, 3.55, 3.02, 7.22, for which the average is 5.44. Fréchet Inception Distance (FID) computed from 50K images was 7.30. See the video for latent space interpolations.

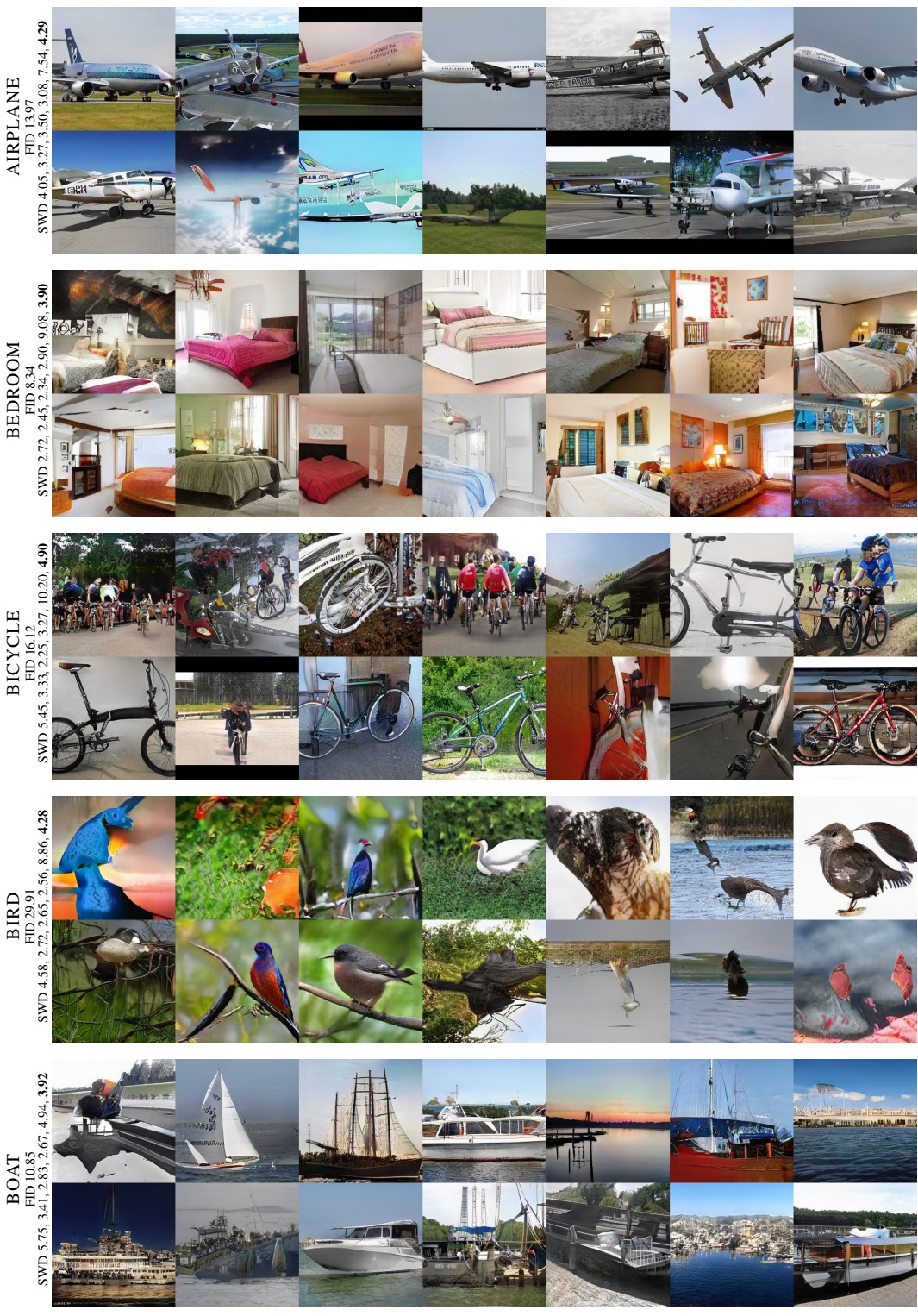

Figure 12: Example images generated at $256 \times 256$ from LSUN categories. Sliced Wasserstein Distance (SWD) $\times 10^3$ is given for levels 256, 128, 64, 32 and 16, and the average is bolded. We also quote the Fréchet Inception Distance (FID) computed from 50K images.

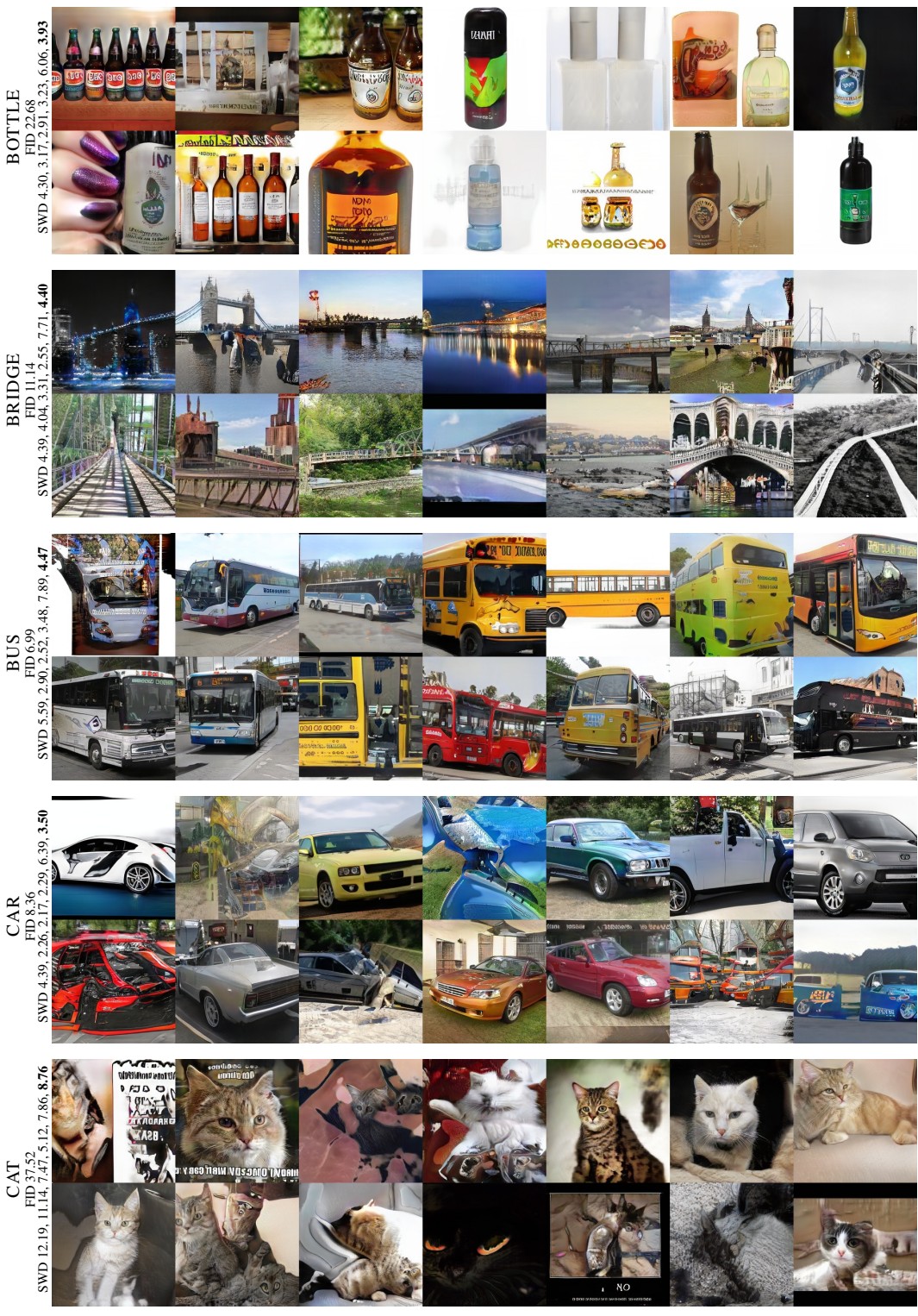

Figure 13: Example images generated at $256 \times 256$ from LSUN categories. Sliced Wasserstein Distance (SWD) $\times 10^3$ is given for levels 256, 128, 64, 32 and 16, and the average is bolded. We also quote the Fréchet Inception Distance (FID) computed from 50K images.

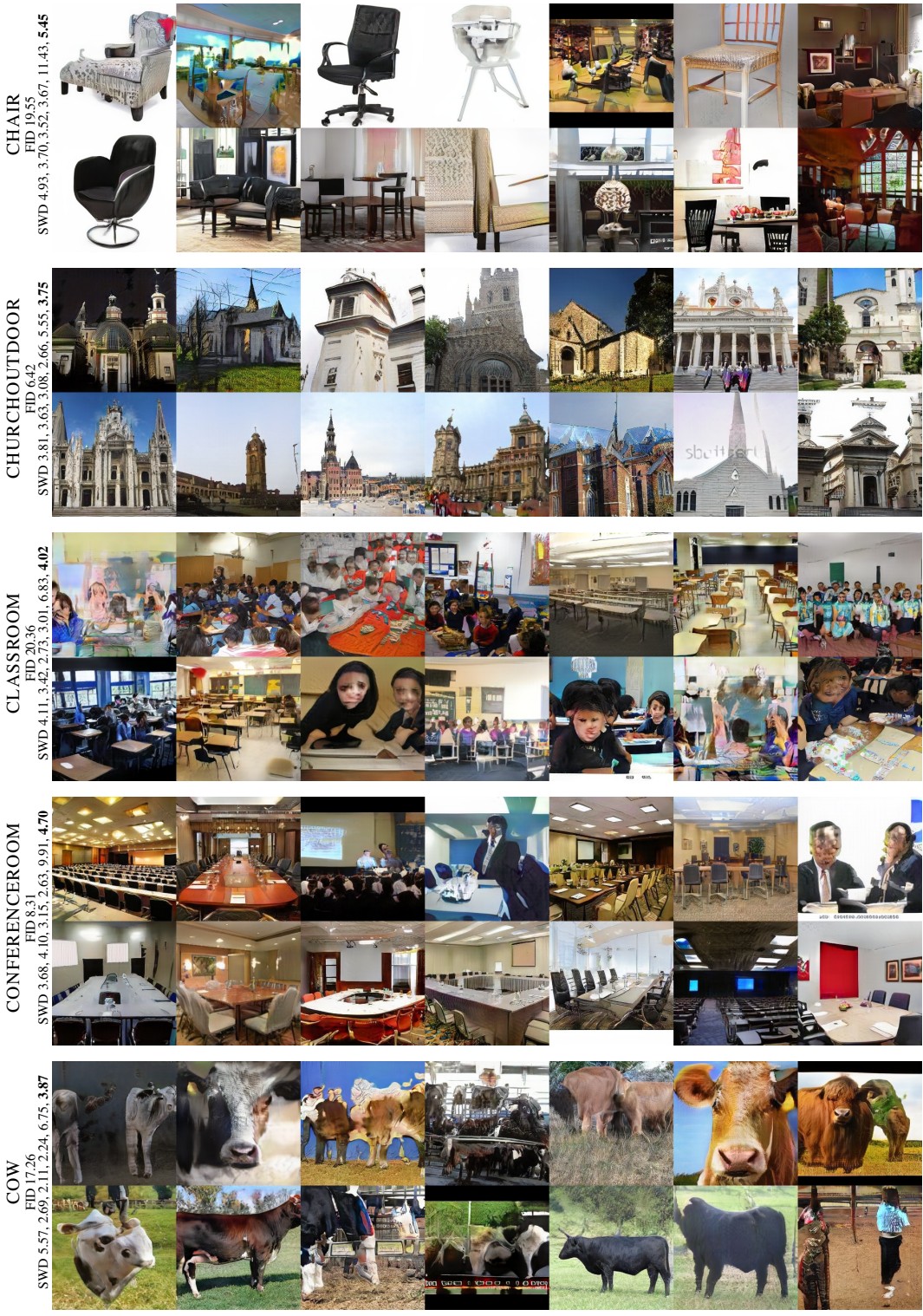

Figure 14: Example images generated at $256 \times 256$ from LSUN categories. Sliced Wasserstein Distance (SWD) $\times 10^3$ is given for levels 256, 128, 64, 32 and 16, and the average is bolded. We also quote the Fréchet Inception Distance (FID) computed from 50K images.

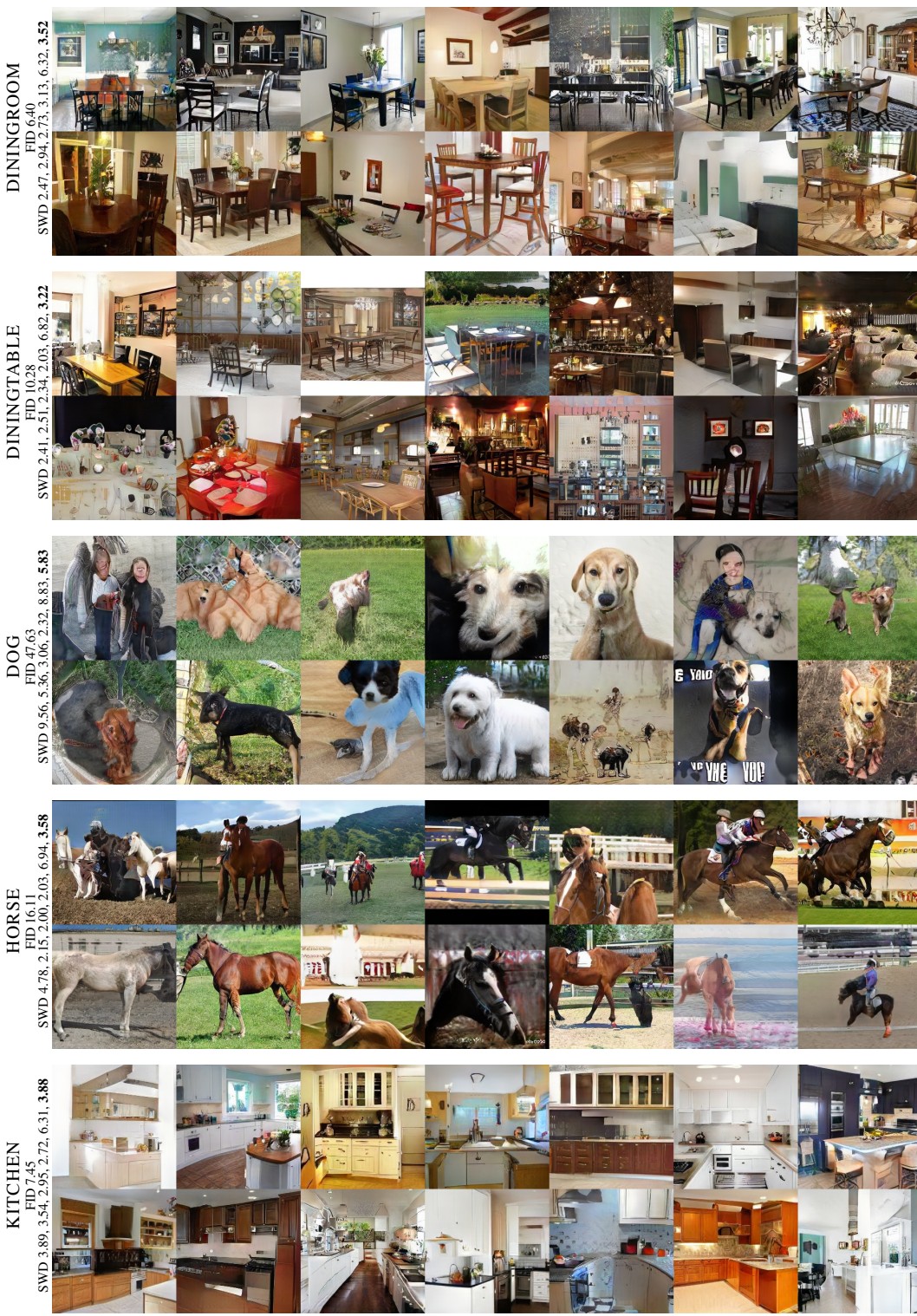

Figure 15: Example images generated at $256 \times 256$ from LSUN categories. Sliced Wasserstein Distance (SWD) $\times 10^3$ is given for levels 256, 128, 64, 32 and 16, and the average is bolded. We also quote the Fréchet Inception Distance (FID) computed from 50K images.

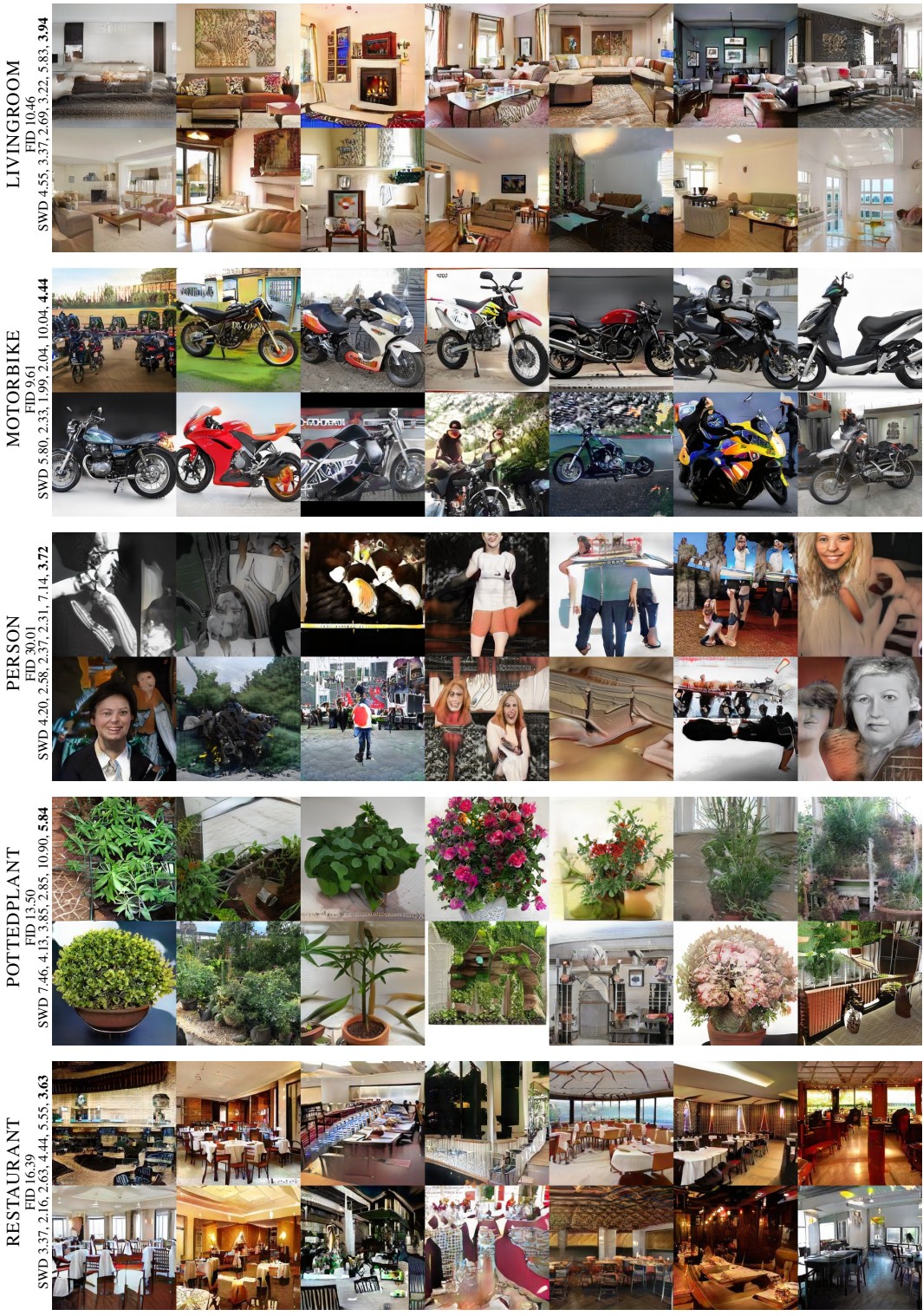

Figure 16: Example images generated at $256 \times 256$ from LSUN categories. Sliced Wasserstein Distance (SWD) $\times 10^3$ is given for levels 256, 128, 64, 32 and 16, and the average is bolded. We also quote the Fréchet Inception Distance (FID) computed from 50K images.

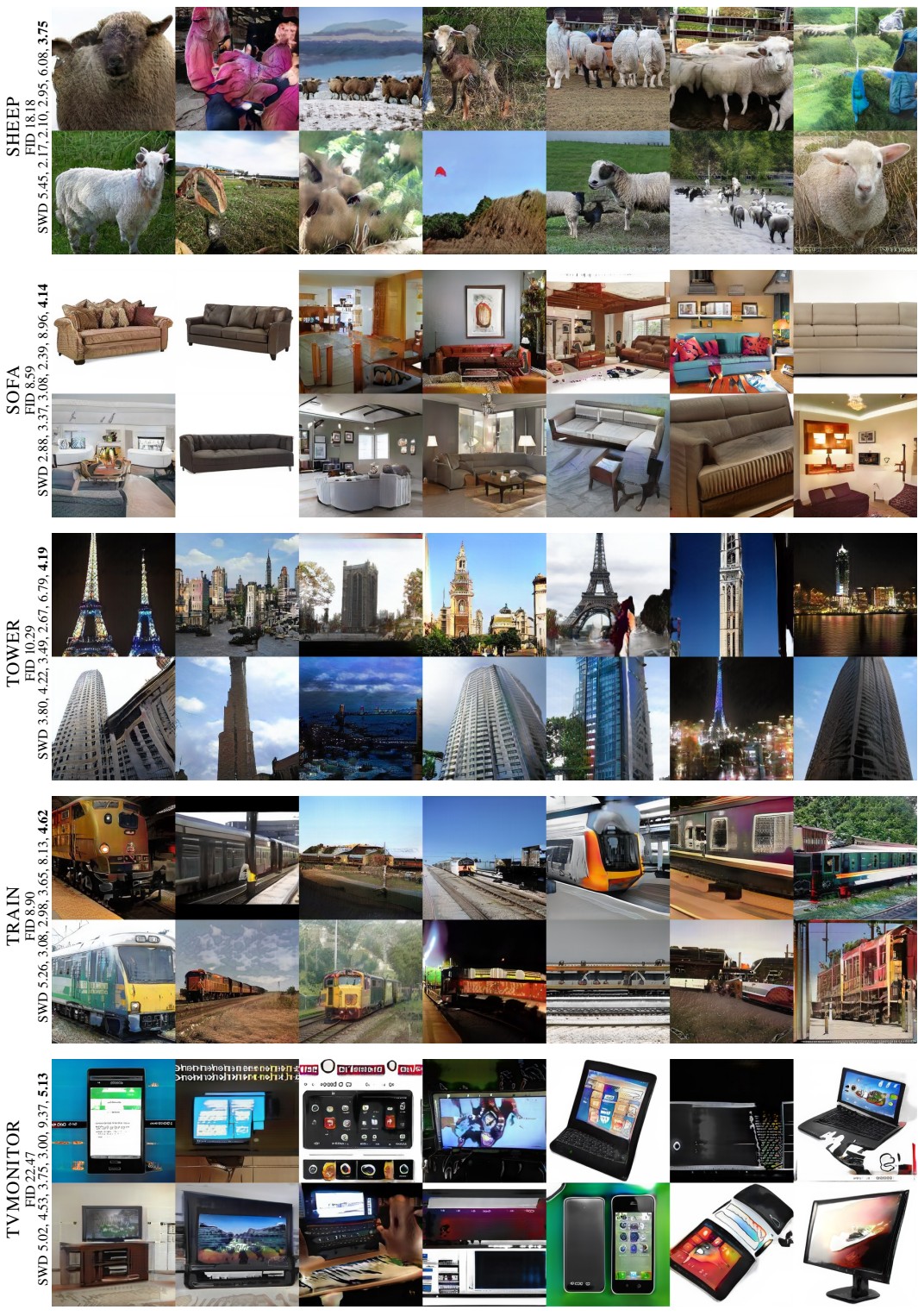

Figure 17: Example images generated at $256 \times 256$ from LSUN categories. Sliced Wasserstein Distance (SWD) $\times 10^3$ is given for levels 256, 128, 64, 32 and 16, and the average is bolded. We also quote the Fréchet Inception Distance (FID) computed from 50K images.

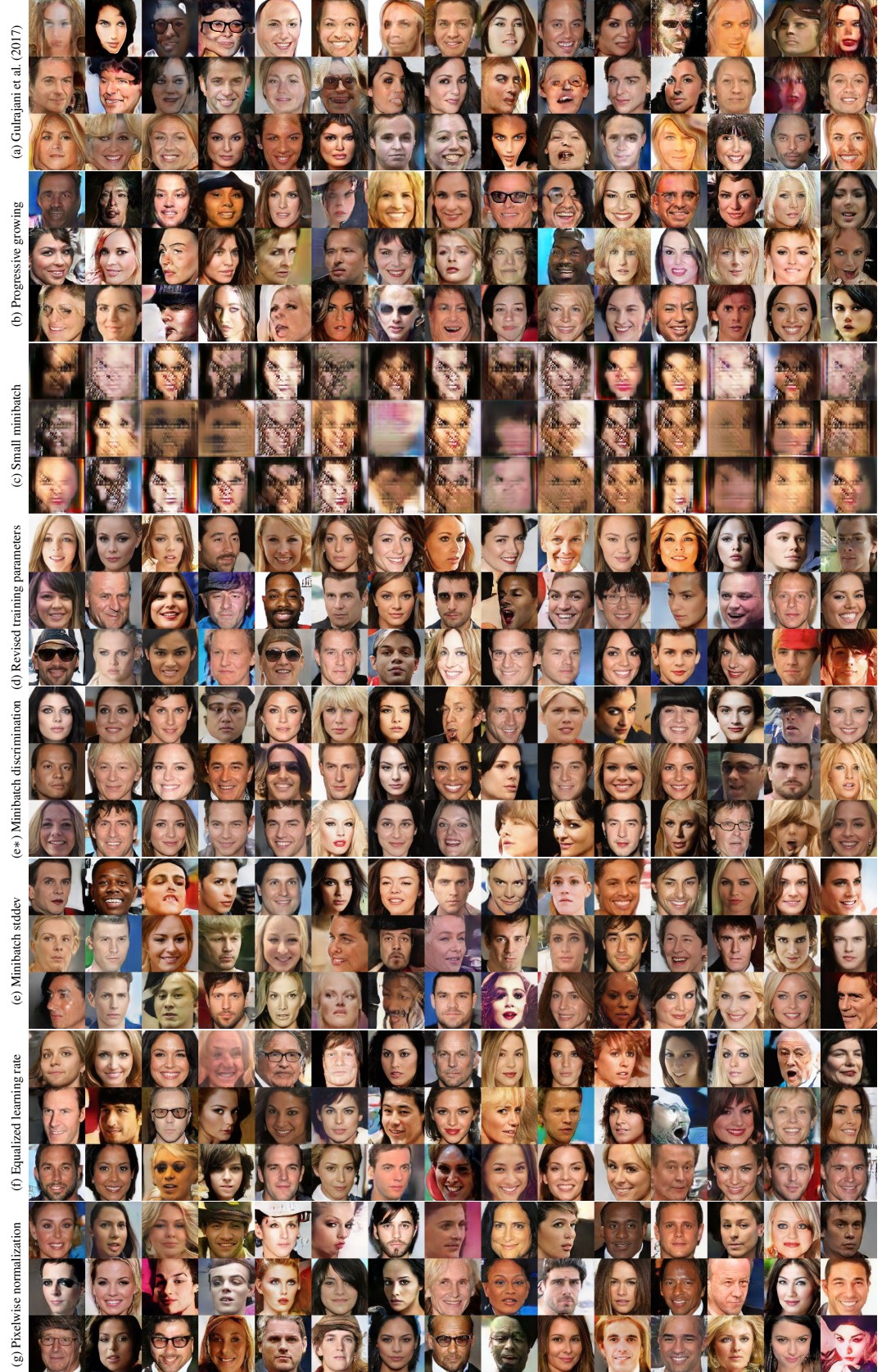

Figure 18: A larger set of generated images corresponding to the non-converged setups in Table 1.

