# OpenReview forum: "Progressive Growing of GANs for Improved Quality, Stability, and Variation"
_ICLR.cc/2018/Conference — Accept (Oral)_

### Official Review · AnonReviewer3 · 2017-11-20
**good paper, accept**

**Rating:** 8
**Confidence:** 4

**Review:**

The paper describes a number of modifications of GAN training that enable synthesis of high-resolution images. The modifications also support more automated longer-term training, and increasing variability in the results.

The key modification is progressive growing. First, a GAN is trained for image synthesis at very low resolution. Then a layer that refines the resolution is progressively faded in. (More accurately, a corresponding pair of layers, one in the generation and one in the discriminator.) This progressive fading in of layers is repeated, one octave at a time, until the desired resolution is reached.

Another modification reported in the paper is a simple parameter-free minibatch summary statistic feature that is reported to increase variation. Finally, the paper describes simple schemes for initialization and feature normalization that are reported to be more effective than commonly used initializers and batchnorm.

It's a very nice paper. It does share the "bag of tricks" nature of many GAN papers, but as such it is better than most of the lot. I appreciate that some of the tricks actually simplify training, and most are conceptually reasonable. The paper is also very well written.

My quibbles are minor. First, I would discuss [Huang et al., CVPR 2017] and the following paper more prominently:

[Zhang et al., ICCV 2017] H. Zhang, T. Xu, H. Li, S. Zhang, X. Wang, X. Huang, and D. Metaxas. StackGAN: Text to photo-realistic image synthesis with stacked generative adversarial networks. In ICCV, 2017.

I couldn't find a discussion of [Huang et al., CVPR 2017] at all, although it's in the bibliography. (Perhaps I overlooked the discussion.) And [Zhang et al., ICCV 2017] is quite closely related, since it also tackles high-resolution synthesis via multi-scale refinement. These papers don't diminish the submission, but they should be clearly acknowledged and the contribution of the submission relative to these prior works should be discussed.

Also, [Rabin et al., 2011] is cited in Section 5 but I couldn't find it in the bibliography.

---

### Official Review · AnonReviewer1 · 2017-11-25
**Mixed - great results on image generation, but not properly anonymized**

**Rating:** 1
**Confidence:** 4

**Review:**

Before the actual review I must mention that the authors provide links in the paper that immediately disclose their identity (for instance, the github link). This is a violation of double-blindness, and in any established double-blind conference this would be a clear reason for automatic rejection. In case of ICLR, double-blindness is new and not very well described in the call for papers, so I guess it’s up to ACs/PCs to decide. I would vote for rejection. I understand in the age of arxiv and social media double-blindness is often violated in some way, but here the authors do not seem to care at all.

—

The paper proposes a collections of techniques for improving the performance of Generative Adversarial Networks (GANs). The key contribution is a principled multi-scale approach, where in the process of training both the generator and the discriminator are made progressively deeper and operate on progressively larger images. The proposed version of GANs allows generating images of high resolution (up to 1024x1024) and high visual quality.

Pros:
1) The visual quality of the results is very good, both on faces and on objects from the LSUN dataset. This is a large and clear improvement compared to existing GANs.
2) The authors perform a thorough quantitative evaluation, demonstrating the value of the proposed approach. They also introduce a new metric - Sliced Wasserstein Distance.
3) The authors perform an ablation study illustrating the value of each of the proposed modifications.

Cons:
1) The paper only shows results on image generation from random noise. The evaluation of this task is notoriously difficult, up to impossible (Theis et al., ICLR 2016). The authors put lots of effort in the evaluation, but still:
- it is unclear what is the average quality of the samples - a human study might help
- it is unclear to which extent the images are copied from the training set.  The authors show some nearest neighbors from the training set, but very few and in the pixel space, which is known to be pointless (again, Theis et al. 2016). Interpolations in the latent space is a good experiment, but in fact the interpolations do not look that great on LSUN
- it is unclear if the model covers the full diversity of images (mode collapse)
It would be more convincing to demonstrate some practical results, for instance inpainting, superresolution, unsupervised or semi-supervised learning, etc.
2) The general idea of multi-scale generation is not new, and has been investigated for instance in LapGAN (Denton et al., ICLR 2015) or StackGAN (Zhang et al., ICCV2017, arxiv 2017). The authors should properly discuss this.
3) The authors mention “unhealthy competition” between the discriminator and the generator several times, but it is not quite clear what exactly they mean - a more specific definition would be useful.

(This conclusion does not take the anonymity violation into account. Because of the violation I believe the paper should be rejected. Of course I am open to discussions with ACs/PCs.)
To conclude, the paper demonstrates a breakthrough in the quality and resolution of images generated with a GAN. The experimental evaluation is thorough, to the degree allowed by the poorly defined task of generating images from random noise. Results on some downstream tasks, such as inpainting, image processing or un-/semi-supervised learning would make the paper more convincing. Still, the paper should definitely be accepted for publication. Normally, I would give the paper a rating of 8.

---

### Official Review · AnonReviewer2 · 2017-11-30
**-**

**Rating:** 8
**Confidence:** 4

**Review:**

This paper proposes a number of ideas for improving GANs for image generation, highlighting in particular a curriculum learning strategy to progressively increase the resolution of the generated images, resulting in GAN generators capable of producing samples with unprecedented resolution and visual fidelity.


Pros:

The paper is well-written and the results speak for themselves! Qualitatively they’re an impressive and significant improvement over previous results from GANs and other generative models.  The latent space interpolations shown in the video (especially on CelebA-HQ) demonstrate that the generator can smoothly transition between modes and convince me that it isn’t simply memorizing the training data. (Though I think this issue could be addressed a bit better -- see below.) Though quantification of GAN performance is difficult and rapidly evolving, there is a lot of quantitative analysis all pointing to significant improvements over previous methods.

A number of new tricks are proposed, with the ablation study (tab 1 + fig 3) and learning curves (fig 4) giving insight into their effects on performance.  Though the field is moving quickly, I expect that several of these tricks will be broadly adopted in future work at least in the short to medium term.

The training code and data are released.


Cons/Suggestions:

It would be nice to see overfitting addressed and quantified in some way.  For example, the proposed SWD metric could be recomputed both for the training and for a held-out validation/test set, with the difference between the two scores measuring the degree of overfitting.  Similarly, Danihelka et al. [1] show that an independently trained Wasserstein critic (with one critic trained on G samples vs. train samples, and another trained on G samples vs. val samples) can be used to measure overfitting.  Another way to go could be to generate a large number of samples and show the nearest neighbor for a few training set samples and for a few val set samples.  Doing this in pixel space may not work well especially at the higher resolutions, but maybe a distance function in the space of some high-level hidden layer of a trained discriminator could show good semantic nearest neighbors.

The proposed SWD metric is interesting and computationally convenient, but it’s not clear to me that it’s an improvement over previous metrics like the independent Wasserstein critic proposed in [1].  In particular the use of 7x7 patches would seem to limit the metric’s ability to capture the extent to which global structure has been learned, even though the patches are extracted at multiple levels of the Laplacian pyramid.

The ablation study (tab 1 + fig 3) leaves me somewhat unsure which tricks contribute the most to the final performance improvement over previous work.  Visually, the biggest individual improvement is easily when going from (c) to (d), which adds the “Revised training parameters”, with the improvement from (a) to (b) which adds the highlighted progressive training schedule appearing relatively minor in comparison.  However, I realize the former large improvement is due to the arbitrary ordering of the additions in the ablation study, with the small minibatch addition in (c) crippling results on its own.  Ablation studies with large numbers of tweaks are always difficult and this one is welcome and useful despite the ambiguity.

On a related note, it would be nice if there were more details on the “revised training hyperparameters” improvement ((d) in the ablation study) -- which training hyperparameters are adjusted, and how?

“LAPGAN” (Denton et al., 2015) should be cited as related work: LAPGAN’s idea of using a separate generator/discriminator at each level of a Laplacian pyramid conditioned on the previous level is quite relevant to the progressive training idea proposed here.  Currently the paper is only incorrectly cited as “DCGAN” in a results table -- this should be fixed as well.


Overall, this is a well-written paper with striking results and a solid effort to analyze, ablate, and quantify the effect of each of the many new techniques proposed. It’s likely that the paper will have a lot of impact on future GAN work.


[1] Danihelka et al., “Comparison of Maximum Likelihood and GAN-based training of Real NVPs” https://arxiv.org/abs/1705.05263

---

### Public Comment · (anonymous) · 2017-11-12
**Questions**

1. Was performance degrading actually observed when smoothing was not used?

2. It seems that the reported Inception score of CIFAR is based on minibatch size of 16 because you wanted to show that the performance degrading of small minibatch could be remedied by that. Have you found a significant increase in score when the size is 64 and when you use all the techniques you used to remedy the aforementioned performance degrading? If so, I think the degree of the increase in score would indicate the size of this bottleneck.

3. Did you do any other architecture exploration such as beginning from first block having 2x2 output instead of 4x4?

---

> ### Author Response · Authors · 2017-11-15
> **Answers**
>
> 1.
> Yes, assuming that the question refers to the exponential running average that we use for visualizing the generated images.
>
> We have observed that the best results are generally obtained using a relatively high learning rate, which in turn leads to significant variation in terms of network weights between consecutive training iterations. Any instantaneous snapshot of the generator is likely to be slightly off or exaggerated in terms of various image features such as color, brightness, sharpness, shape of the mouth, amount of hair, color of the eyes, etc. The exponential running average reduces this variation, leading to considerably more consistent results.
>
> Intuitively speaking, we can say that the generator and discriminator are constantly exploring a large neighborhood of different solutions around the current average solution, even though the average solution itself evolves relatively slowly. According to our experiments, such exploration seems to be highly beneficial in terms of eventually converging towards a good local optimum.
>
> 2.
> We have tried increasing the minibatch size in our CIFAR-10 runs, but we have not observed an increase in the inception scores.
>
> The performance degradation associated with small minibatch size is largely limited to configurations that rely heavily on batch normalization (rows a-c in Table 1). Perhaps surprisingly, we have observed that smaller minibatches actually produce slightly better results in configurations where batch normalization is not present (rows d-h).
>
> 3.
> We did explore different network architectures in the early stages of the project. In general, it does not seem to make a big difference whether we start at 2x2, 4x4, 8x8, or 16x16 resolution. We chose 4x4 mainly because it is the most natural fit for our specific network architecture. We have also observed that it is beneficial to have roughly the same structure and capacity in both networks, as well as matching upsampling and downsampling operators.

---

### Public Comment · (anonymous) · 2017-11-29
**Minor Correction**

In Section 4.1 you mention that you are initializing the network weights by sampling from the normal distribution. In your code, it appears you are using the stock Lasagne weight initialization, which uses the Xavier Glorot uniform distribution. Or has this changed in some newer version of Lasagne?

---

> ### Author Response · Authors · 2017-11-29
> **Re: Minor Correction**
>
> We apologize that the source code is somewhat convoluted in this regard.
>
> Our implementation performs weight initialization in two distinct phases. It first initializes the weights using Lasagne's standard He initializer and then rescales them in accordance to Section 4.1. For example, consider the following line in network.py (http://bit.ly/2zykV3P#L471):
>
> 471  net = PN(BN(WS(Conv2DLayer(net, name='G1b', num_filters=nf(1), filter_size=3, pad=1, nonlinearity=act, W=iact))))
>
> Here, we create a standard Conv2DLayer and apply equalized learning rate (WS) as well as pixelwise feature vector normalization (PN) on top of it. Note that batch normalization (BN) is disabled in most of our experiments. When the Conv2DLayer is first instantiated, the weights are initialized according to W=iact, which in turn is defined as lasagne.init.HeNormal('relu') on lines 459 and 32. We apply equalized learning rate by latching a custom WScaleLayer (line 278) onto the Conv2DLayer. When the WScaleLayer is instantiated, it estimates the elementwise standard deviation of the weights and normalizes them accordingly:
>
> 281  W = incoming.W.get_value()
> 282  scale = np.sqrt(np.mean(W ** 2))
> 283  incoming.W.set_value(W / scale)
>
> The value on line 281 corresponds to $\hat{w}_i$ in the paper, line 282 corresponds to $c$, and line 283 to $w_i$. In other words, this part of the code undoes the effect of He's initializer and brings the weights back to trivial N(0,1) initialization.

---

### Author Response · Authors · 2017-12-05
**We thank the reviewers for accurate reviews**

We will fix all the references and add related discussion to the paper.

We have, in fact, obtained more sensible nearest-neighbor results using a feature-space distance metric for image comparison. We will update Fig. 10 accordingly and include multiple nearest neighbors for each generated image. The conclusion still stands that the generated images have no obvious source images in the training set. The hyperparameter changes related to Table 1 (d) are listed in Appendix A.2.

We acknowledge R1’s concerns about anonymity and feel a few words are in order.

First, the call for papers explicitly states that arXiv and other such public forums are permitted. While we agree that full anonymity is valuable, we feel that one cannot realistically expect to achieve it perfectly, because so many potential reviewers subscribe to the arXiv announce list and articles from that list are inevitably discussed in social media.

Second, the OpenReview submission site does not allow supplemental videos or code, forcing one to use services like YouTube and GitHub, neither of which allows anonymous submissions. In our opinion, that leaves two possibilities: 1) fake accounts, or 2) breach of anonymity. We thought about this long and hard and chose #2 because #1 seems fraught with many more problems -- and would perhaps also seem like a strange requirement. While we anonymized the paper and the video to the extent possible within these bounds, we regrettably forgot a full author list in the readme at GitHub. We sincerely apologize for this oversight.

In order to avoid this kind of awkwardness in the future, we feel that explicit guidance in the CFP -- including suggested best practices for submitting videos, code, and data -- would be helpful and greatly facilitate the review process.

---

> ### Comment · AnonReviewer1 · 2018-01-10
> **Anonymity**
>
> I totally agree with the last point: it would have been great if the organizers provided a more detailed CFP and recommended best practices.
>
> However, I disagree with the other two points:
> First, I know arxiv, talks, blogs, etc are permitted. But directly linking the author list from the paper is generally not.
> Second, there are ways to host data anonymously and many ICLR authors (including myself) found some. If in doubt, the right way would be to ask the organizers, not breach the anonymity directly.
>
> In the end, the decision is on ACs and PCs.

---

> > ### Public Comment · (anonymous) · 2018-05-14
> > **Use anonymous links**
> >
> > You can generate anonymous URLs. For example, dropbox directly lets you do that!

---

### Author Response · Authors · 2017-12-14
**Revised PDF**

We have uploaded a new revision of the paper, addressing the concerns brought up in the reviews. The detailed list of changes is as follows:

- Revise the nearest neighbors in Figure 10 by using VGG feature-space distance and showing 5 best matches for each generated image.
- Report average CIFAR10 inception score over 10 random initializations in Table 3, in addition to the highest achieved score.
- Add discussion of [Denton et al. 2015], [Huang et al. 2016], and [Zhang et al. 2017] in Section 2.
- Fix [Anonymous 2017], [Rabin et al. 2011], and [Radford et al. 2015] references.
- Update "(h) Converged" case in Table 1 and Figure 3, as well as LSUN images in Figures 12-17 using networks that were trained longer.
- Report SWD numbers for CelebA-HQ and LSUN categories in Figures 11-17.
- Typo fixes and minor clarifications.

We would like to thank the reviewers again for useful feedback.

---

### Author Response · Authors · 2018-02-23
**Camera ready version**

We have uploaded the de-anonymized camera ready version of the paper.

---

### Public Comment · ~Marek_Krcal1 · 2018-05-14
**Birthday paradox test**

The work seems very impressive to me (but I am a layman). Have you tried the birthday paradox test (Arora et al.) to measure the support size of your generator's distribution?

---

> ### Author Response · Authors · 2018-05-15
> **Yes**
>
> We did a few tests with it early on, but were not able to find obvious duplicate images especially in higher resolutions. The test is a clever idea but it's also highly subjective and thus quite difficult to use in practice.

---

### Decision · Program_Chairs · 2018-01-29
**ICLR 2018 Conference Acceptance Decision**

**Decision:**

Accept (Oral)

**Comment:**

The main contribution of the paper is a technique for training GANs which consists in progressively increasing the resolution of generated images by gradually enabling layers in the generator and the discriminator. The method is novel, and outperforms the state of the art in adversarial image generation both quantitatively and qualitatively. The evaluation is carried out on several datasets; it also contains an ablation study showing the effect of contributions (I recommend that the authors follow the suggestions of AnonReviewer2 and further improve it). Finally, the source code is released which should facilitate the reproducibility of the results and further progress in the field.

AnonReviewer1 has noted that the authors have revealed their names through GitHub, thus violating the double-blind submission requirement of ICLR; if not for this issue, the reviewer’s rating would have been 8. While these concerns should be taken very seriously, I believe that in this particular case the paper should still be accepted for the following reasons:
1) the double blind rule is new for ICLR this year, and posting the paper on arxiv is allowed;
2) the author list has been revealed through the supplementary material (Github page) rather than the paper itself;
3) all reviewers agree on the high impact of the paper, so having it presented and discussed at the conference would be very useful for the community.